# Resurgence of Omicron BA.2 in SARS-CoV-2 infection-naive Hong Kong

Ruopeng Xie[1,2,6], Kimberly M. Edwards [1,2,6], Dillon C. Adam[1], Kathy S. M. Leung [1], Tim K. Tsang [1], Shreya Gurung[1,2], Weijia Xiong[1], Xiaoman Wei[1,2], Daisy Y. M. Ng[1], Gigi Y. Z. Liu[1], Pavithra Krishnan[1], Lydia D. J. Chang[1], Samuel M. S. Cheng[1], Haogao Gu [1], Gilman K. H. Siu[3], Joseph T. Wu [1,4], Gabriel M. Leung [1,4], Malik Peiris [1,5], Benjamin J. Cowling [1,4], Leo L. M. Poon [1,2,5] & Vijaykrishna Dhanasekaran [1,2] ✉

Hong Kong experienced a surge of Omicron BA.2 infections in early 2022, resulting in one of the highest per-capita death rates of COVID-19. The outbreak occurred in a dense population with low immunity towards natural SARS-CoV-2 infection, high vaccine hesitancy in vulnerable populations, comprehensive disease surveillance and the capacity for stringent public health and social measures (PHSMs). By analyzing genome sequences and epidemiological data, we reconstructed the epidemic trajectory of BA.2 wave and found that the initial BA.2 community transmission emerged from cross-infection within hotel quarantine. The rapid implementation of PHSMs suppressed early epidemic growth but the effective reproduction number ($R_e$) increased again during the Spring festival in early February and remained around 1 until early April. Independent estimates of point prevalence and incidence using phylodynamics also showed extensive superspreading at this time, which likely contributed to the rapid expansion of the epidemic. Discordant inferences based on genomic and epidemiological data underscore the need for research to improve near real-time epidemic growth estimates by combining multiple disparate data sources to better inform outbreak response policy.

After the initial global spread of SARS-CoV-2 in 2020, new waves of infection have been triggered by the emergence of novel Variants of Concern (VOC) such as Alpha, Beta, Gamma, Delta, and most recently, Omicron and its subvariants with greater transmissibility, significant immune evasion and capacity for strong vaccine breakthrough[1]. In response to more contagious variants, countries that maintained elimination strategies throughout 2021, such as New Zealand and Singapore, pivoted towards mitigation[2]. However, Hong Kong (population. 7.4 million), which successfully eliminated four distinct waves of sustained SARS-CoV-2 transmission between January 2020 to April 2021, continued to maintain its elimination policy into early 2022. In January 2022, Hong Kong experienced a surge of SARS-CoV-2 Omicron subvariant infections that quickly overwhelmed the health care system, isolation facilities, and track-and-trace capacities (Fig. 1). Between

[1]School of Public Health, LKS Faculty of Medicine, The University of Hong Kong, Hong Kong SAR, China. [2]HKU-Pasteur Research Pole, School of Public Health, LKS Faculty of Medicine, The University of Hong Kong, Hong Kong SAR, China. [3]Department of Health Technology and Informatics, The Hong Kong Polytechnic University, Hong Kong SAR, China. [4]Laboratory of Data Discovery for Health, Hong Kong Science and Technology Park, New Territories, Hong Kong SAR, China. [5]Centre for Immunology & Infection, Hong Kong Science and Technology Park, New Territories, Hong Kong SAR, China. [6]These authors contributed equally: Ruopeng Xie, Kimberly M. Edwards. ✉e-mail: veej@hku.hk

**Fig. 1 | Epidemiological summary of SARS-CoV-2 from January to April 2022 in Hong Kong.** Reported cases and deaths (above) and sequenced genomes (below) over time. Rapid antigen test-positive cases reported on 9 March include cases from both 8 and 9 March.

February–April 2022, Hong Kong saw one of the highest COVID-19 per-capita death rates among high-income countries, with over 9000 deaths in these three months (peak of 3.5 per 100,000 people per day) compared to just 213 cumulative deaths in the preceding two years. Deaths were disproportionately attributed to older adults (65+ years), many of whom were unvaccinated[3,4]. Due to the low vaccine coverage in this population[5], residential care homes for the elderly and disabled were significantly affected. Even mild cases in these settings resulted in increased morbidity due to disruption of normal care. As established systems for testing became overwhelmed, the Centre for Health Protection (CHP) pivoted to include positive rapid antigen test (RAT) cases from private hospitals and laboratories in official case counts since 26 February (Fig. 1), rather than only recognizing PCR-positives confirmed by government reference laboratories (Fig. 1). A self-declaration system for positive RAT reporting was launched on 7 March. Amidst various changes in case counting strategies and the sudden overload of the testing system, it is likely that the true incidence of COVID-19 cases during this period was substantially underreported.

In contrast to other elimination-focused countries, Omicron's emergence in Hong Kong occurred in a context of reduced population immunity to SARS-CoV-2 due to the effectiveness of past suppression measures and therefore limited prior infection rates, as well as low vaccination rates among high-risk populations[6,7]. Furthermore, a recent survey showed that medical misinformation, political distrust, and complacency (especially among the elderly) borne from a lowered risk perception (given the effective control of the pandemic in Hong Kong thus far), substantially contributed to this lowered incidence of "hybrid population immunity" (i.e., infection and vaccine-acquired immunity) by the end of 2021[8].

A wide range of public health and social measures (PHSMs) were already in place at the start of 2022, including universal masking, travel restrictions, an app-based "Leave Home Safe" track-and-trace system, and limits on social gathering and dining. In response to reports of the emergence of the Omicron subvariant, high-risk gatherings were pre-emptively restricted, including the complete closure of entertainment venues such as bars and the closure of dine-in venues between 6 pm–5 am (Fig. 1). Furthermore, persons who had visited countries perceived as high-risk were temporally banned from entering Hong Kong, with direct flight routes from these countries also banned.

Face-to-face teaching for primary levels was suspended on 14 January, and for secondary schools on 23 January 2022 (Fig. 1). However, restrictions on social gatherings were later relaxed during the Chinese New Year (Spring Festival) between 1 and 3 February.

In this study, we combine epidemiological records and 3317 genome sequences collected during the fifth SARS-CoV-2 wave in Hong Kong (January to April 2022) to reveal the epidemic and evolutionary trajectory of circulating variants across a densely populated and largely infection-naive population under strict PHSMs. We also provide an independent estimate of the cumulative incidence of BA.2.2 infection that does not rely on case counts.

## Results

### Genomic epidemiology of the fifth wave in Hong Kong

Daily locally reported cases for the population of 7.4 million remained below 20 until 21 January 2022 and below 500 until 6 February. Daily cases increased gradually to around 10,000 on 25 February, followed by >50,000 daily cases for eight days from 26 February to 4 March, peaking at >70,000 cases on four of these days. The sharp rise in cases in late February reflects the inclusion of rapid antigen tests (RAT), which accounted for 36% of reported cases during the peak (>20,000 cases) from 26 February to 17 March 2022. Cases declined from mid-March and throughout April, from ~20,000 cases on 18 March to <500 cases per day on 24 April 2022 (Fig. 1). In the first four months of 2022, 9095 COVID-19 deaths were reported in Hong Kong. Similar to the first four COVID-19 waves in Hong Kong, local outbreaks clustered in areas of high population density (Supplementary Fig. 1a). Across the 18 districts of Hong Kong, COVID-19 incidence from January to March 2022 was negatively correlated with median income (Spearman's rank correlation, rho ($\rho$) = −0.81, $p < 0.001$) and positively correlated with population density (rho ($\rho$) = 0.48, $p = 0.047$) (Supplementary Fig. 1b). In contrast, the incidence of imported cases from January 2020 to January 2021 was positively correlated with median income (Spearman's rank correlation, rho ($\rho$) = 0.56, $p = 0.016$) (Supplementary Fig. 1b).

Hong Kong's fifth wave commenced with the detection of multiple SARS-CoV-2 VOC in the community (Figs. 1 and 2a). Based on genome sequencing, most COVID-19 cases from January to April 2022 ($n = 3317$) were caused by Omicron BA.2 and related sublineages (BA.2*) ($n = 2807$; 85%), while Omicron BA.1* ($n = 383$) and Delta AY.127

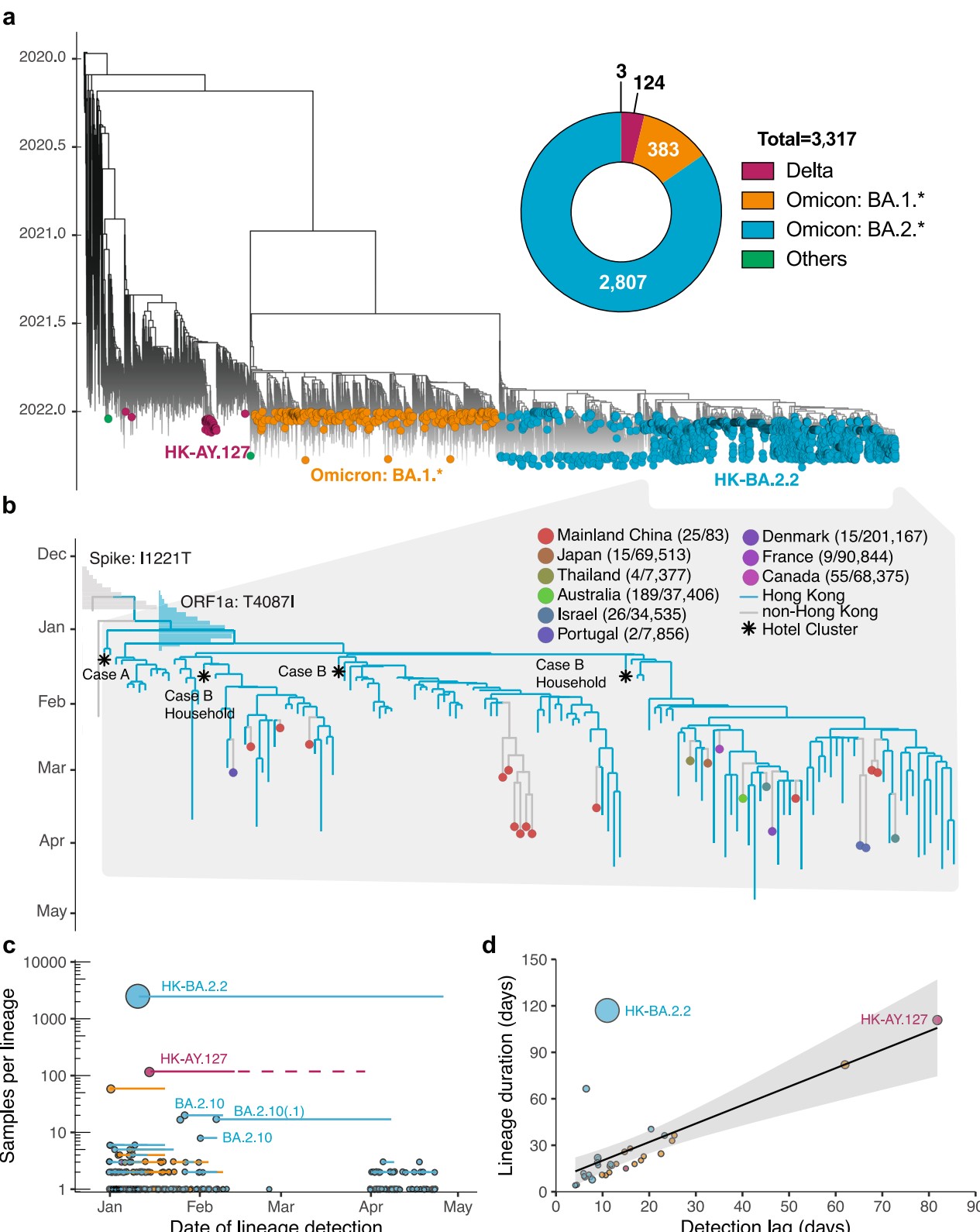

(n = 126) lineages were detected in limited numbers. The majority of BA.1* samples were detected in January from travel-related cases with the limited onward transmission. BA.1* formed 252 independent transmission lineages, of which 80.2% did not seed detectable onward transmission, 13.9% resulted in one additional local case (i.e., singleton), and 6.0% led to onward transmissions with durations of less than three weeks. The two largest monophyletic clades (n = 59 and n = 6) were related to a dance cluster[9,10] of 53 cases (some cases were

sequenced more than once) and 16 cases linked to a restaurant cluster[11,12] introduced by flight crew (Fig. 2a and Supplementary Data 1). Four of five Delta introductions in January 2022 were contained within one to two transmission events (Fig. 2a and Supplementary Data 2). Delta cases detected in the community between 15 January and 13 February formed a single monophyletic lineage introduced by imported pet hamsters and first reported on 17 January 2022 (Supplementary Fig. 2)[13,14].

**Fig. 2 | Phylogenetic analysis and transmission lineages of the fifth wave outbreak from January to April 2022 in Hong Kong. a** Time-scaled ML tree of 3317 viruses sampled from Hong Kong (colored circles) on a background of 5220 sub-sampled global viruses collected throughout the pandemic (no tips). **b** Time-resolved maximum clade credibility (MCC) tree of BA.2.2 lineage (*n* = 121 in Hong Kong and *n* = 26 global sequences after subsampling; see Methods). The posterior distribution of the time to the most recent common ancestor (tMRCA) is shown in bar charts. The numbers in the brackets indicate BA.2.2 sequences for each country over the total number of sequences in GISAID (accessed 1 May 2022) from January to April 2022. **c** Summary of SARS-CoV-2 transmission lineages in Hong Kong. Dots represent monophyletic clades colored by lineage and proportional to sequence numbers. Horizontal lines represent the time between the first and last sample dates. After 13 February 2022, there is no AY.127 sequences available in GISAID, but an existing study found the latest AY.127 sequence (incomplete genome) in late March[13], as indicated by the light red dashed line. **d** Correlation between the detection lag (from tMRCA to first sample date) and duration (tMRCA to last sample date). The shaded area represents the fitted values' 95% confidence intervals.

From 2807 BA.2.* sequences sampled from 1 January 2022 to 26 April 2022, onward community transmission was observed in 18 of 214 monophyletic clades (Fig. 2a and Supplementary Data 3); 152 detections did not lead to detectable onward transmission, and 44 were observed as singletons. We identified three BA.2.10 monophyletic clades around February 2022. Among them, two ended quickly, and one, with 14 sequences, was detected from 7 February 2022 to 8 April 2022 and exported to mainland China (Supplementary Fig. 3a). Based on epidemiological records and phylogenetic analysis, this clade originated in Nepal and was repeatedly detected in travel cases (Supplementary Fig. 3a).

The largest monophyletic lineage (HK-BA.2.2 clade, *n* = 2461 sequences) was first detected on 11 January 2022 and most recently sampled on 26 April 2022 (Fig. 2 and Supplementary Data 3). The earliest sequence collected in this lineage was linked to a traveler (Case A) who arrived from Nepal on 4 January 2022 and was quarantined in the Silka Seaview Hotel[15]. This case tested positive on 11 January during quarantine. In the third week of January, BA.2.2 was detected in a community outbreak in a large housing estate[13,15]. Phylogenetic analysis (Fig. 2b and Supplementary Fig. 3b) suggests that Case A infected another inbound traveler in an adjacent room (Case B) who arrived from Pakistan and was soon to be released from a 21-day quarantine (on 10 January 2022). Consequently, Case B tested positive 26 days after arrival in Hong Kong (sampled on 16 January 2022), and thus spread BA.2.2 into the community[13,15,16]. We also found that the HK-BA.2.2 lineage was exported from Hong Kong to at least nine other countries but did not become widespread elsewhere as indicated by a very low proportion (less than 0.5%) of BA.2.2 sequences relative to all sequences, except in mainland China (25/83, ~30%) where insufficient sequences were available (Fig. 2b).

The predominant HK-BA.2.2 lineage contained spike I1221T and ORF1a T4087I substitutions, whereas the ancestral strain traced to Nepal as early as 24 December 2021 contained only the spike I1221T mutation. Bayesian molecular clock analysis showed that the mean time to most recent common ancestor (tMRCA) of viruses with spike I1221T was 19 December 2021 (95% highest posterior density interval (HPD) 8 December 2021 to 24 December 2021), while the mean tMRCA of HK-BA2.2 lineage with both mutations was estimated at 1 January 2022 (HPD, 23 December 2021 to 8 January 2022), substantiating epidemiological findings of BA.2.2 introduction on 4 January 2022 (Fig. 2b).

During the fifth wave, the median delay in detection for non-singleton onward transmission lineages was 11.5 days (95% HPD 4–62 days) (Fig. 2d and Supplementary Data 1–3). A significant correlation (Spearman's test, rho ($\rho$) = 0.72, *p* < 0.001) was found between the lineage detection lag and the lineage duration during the fifth wave, which were similar to those of the first four waves (Spearman's test, rho ($\rho$) = 0.7, *p* < 0.001)[17].

### Dynamics of BA.2.2 lineage

To reveal changes in the spread of BA.2.2 in Hong Kong over time, we used a Bayesian birth-death skyline model that explicitly estimates the rate of transmission, recovery, and sampling, enabling a direct inference of the effective reproduction number ($R_e$) based on sampled sequences and sample dates[18] over 16 time intervals, roughly corresponding to weeks between 3 January and 26 April (Fig. 3a). We observed an increase in $R_e$ to 2.5 (HPD, 1.1–4.2) during the second week (10–16 January 2022), briefly matching the time point on 10 January when Case B left the hotel and introduced the virus into the community. Higher values of $R_e$ (mean, 3.4; HPD, 2.2–4.8) continued to be observed until around 24 January, during the third week. The instantaneous effective reproduction number ($R_t$), estimated from the number of local infections reported per day, increased gradually from 1 (HPD, 0.6–2.2) on 12 January and peaked at 5.2 (HPD, 3.9–7.7) on 20 January. During the third week (17–23 January 2022), $R_e$ was lower than $R_t$, which is most likely due to the co-circulation of multiple lineages (AY.127, BA.1* and BA.2*) (Fig. 1).

$R_e$ decreased to 0.8 (HPD, 0.04–1.9) during the fourth week from 24–30 January 2022, consistent with the suspension of face-to-face teaching for kindergarten and primary schools by 14 January and secondary schools by 22 January, which substantially reduced mobility levels among students in Hong Kong (Supplementary Fig. 5). However, $R_e$ increased again during the fifth week (31 January to 6 February 2022) to 2.7 (HPD, 1.8–3.7) in correlation with a slight increase in mobility levels during the Spring Festival holidays (1–3 February 2022). There was a similar dynamic pattern in $R_t$, but from 7–28 February, $R_t$ remained above 2 with a slight decrease, significantly higher than $R_e$ which fluctuated around 1. Higher $R_t$ may reflect the under-sequencing of HK-BA.2.2 during this period. Especially after 14 February, when infections overwhelmed the health care system, isolation facilities, and track-and-trace capacity, <1% of samples were sequenced (Figs. 1 and 3a and Supplementary Table 1). Interestingly, from March to mid-April, $R_e$ continued to fluctuate around 1 in comparison to $R_t$, which was less than 1, indicating a slower decline of the outbreak than anticipated.

As the rate of coalescence in the phylogeny is proportional to the number of infected individuals during the initial phase of exponential growth[19], we used a Bayesian Skygrid coalescent model[20] to estimate relative changes in effective population size ($N_e$). The early exponential increase in $N_e$ stabilized from late January to early February, coinciding with the decrease in $R_e$. However, $N_e$ rebounded in early February with a sharp increase in late February 2022, peaking around 9 March 2022, and remained relatively stable throughout March (Fig. 3b). Combining $N_e$ with the number of PCR tests conducted and test positivity rate, the Bayes' theorem calculated that the relative case detection rate decreased by ~3–14 fold between 15 January 2022 and 4 February 2022 (Fig. 3c). This inference further confirms underreporting at the start of the fifth wave. Once RATs were incorporated in case counting beginning 26 February 2022 (Fig. 1), the number and positivity rate of PCR tests conducted dropped substantially, leading to the potential for the decrease in relative case detection rate to not accurately reflect reality (Fig. 3c and Supplementary Fig. 6). The sharp rise in $N_e$ coinciding with the inclusion of RAT positives from 26 February (Fig. 3b) suggests BA2.2 sublineages circulating cryptically in the community were better captured when public reporting of RAT positives were included, rather than relying on contact tracing mediated surveillance (Supplementary Fig. 7).

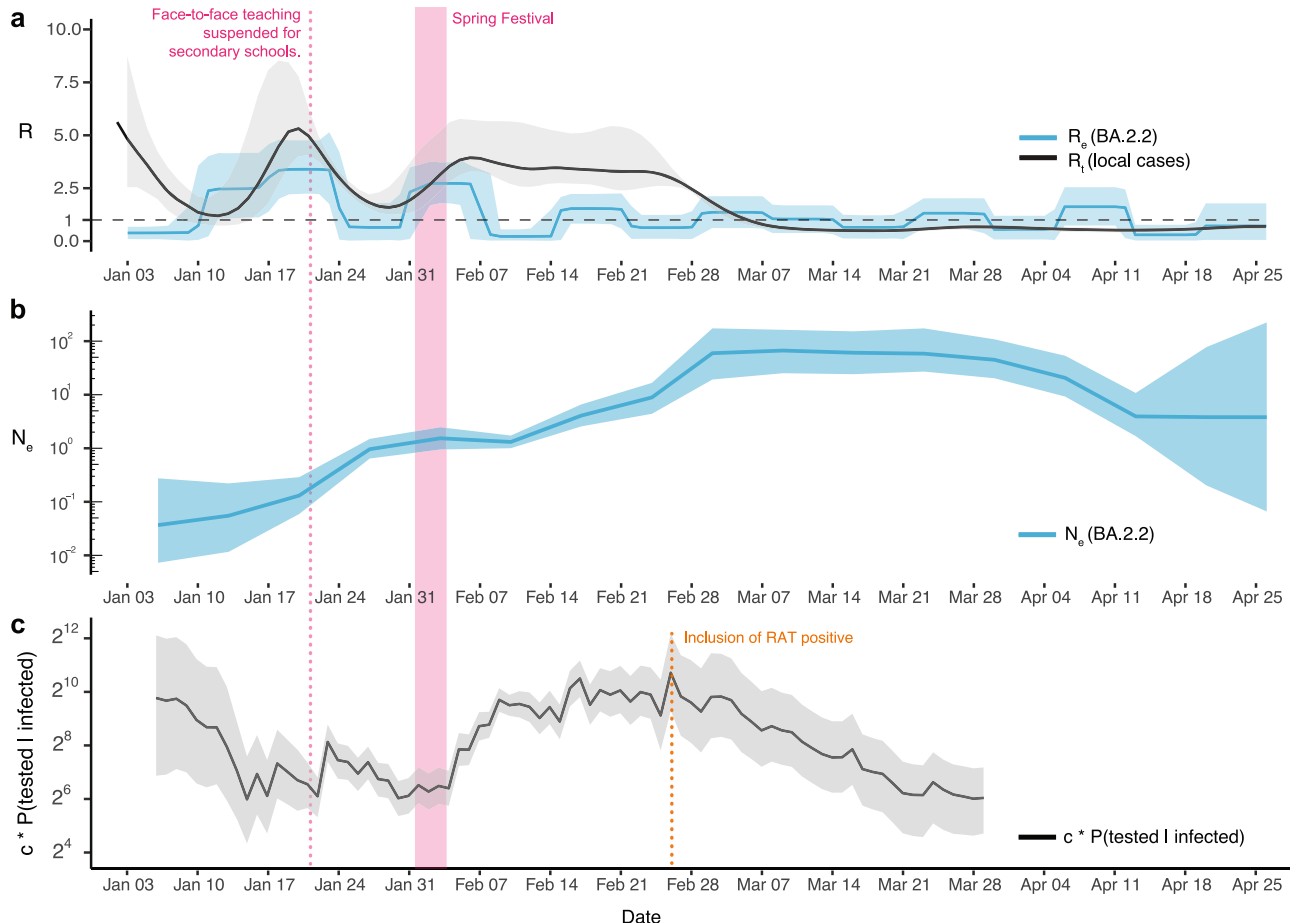

**Fig. 3 | Descriptive dynamics of BA.2.2 lineage in Hong Kong. a** The effective reproduction number ($R_e$) based on BA.2.2 sequences and the instantaneous effective reproduction number ($R_t$) based on the daily reported number of local cases, **b** the effective population size ($N_e$), and **c** relative detection rate in Hong Kong from January 2022 to April 2022. The shaded area denotes the 95% confidence interval.

## An estimation of incidence and prevalence based on levels of superspreading

To better understand the magnitude of BA.2.2 transmission in Hong Kong, we translated $N_e$, estimates using all HK-BA.2.2 genomes from wave five ($n = 2455$), to prevalence ($I$) (see Methods, Fig. 4). We assumed various levels of transmission heterogeneity, a key feature of SARS-CoV-2 transmission[21], measured using the dispersion parameter $k$ ($k = 0.05$, 0.1, 0.15, and 0.2) alongside two levels of generation times $\tau = 2$ or 3 days (Table 1). At the lowest $k = 0.05$, indicating extreme heterogeneity, we estimated 3.55 million infections (95% CI, 1.38–7.40) given $\tau = 2$ days, and 2.23 million infections (95% CI, 0.92–5.85) at $\tau = 3$ days from 6 January 2022 to 11 April 2022 in Hong Kong. In comparison, ~1.18 million cases were officially reported during the same period, indicating an estimated 89 to 184% underreporting rate. At $\tau = 2$ days, we estimated 1.76 million infections (95% CI, 0.72–4.59) given $k = 0.1$ and 1.22 million (95% CI, 0.50–3.20) given $k = 0.15$ (Fig. 4), reducing the rate to 49 and 3% respectively. According to our estimates of prevalence and incidence, the epidemic peaked on the week from 28 Feb to 6 March 2022 (Table 1 and Fig. 4). Despite the inclusion of RAT-positive cases, more substantial under-ascertainment occurred since March, with the exception of under-ascertainment at the start of the fifth wave, as evidenced by a decrease in the relative case detection rate (Fig. 3c).

## Discussion

Under strong border control and community surveillance in Hong Kong during January–April 2022, only two SARS-CoV-2 lineages caused by single introductions circulated (BA.2.2 and AY.127), similar to the pattern observed during the four previous epidemic waves[17]. One BA.2.2 lineage, characterized by an additional ORF1a T4087I mutation, was primarily responsible for the fifth wave and emerged as a result of cross-infection within hotel quarantine. In contrast, we observed a relatively low incidence of AY.127 Delta lineage, linked to an imported hamster-to-human related transmission cluster[14].

We detected an increase in BA.2.2 transmissibility ($R_e = 2.5$; HPD, 1.1–4.2) since 10 January 2022 where the epidemic surge occurred in a densely populated and largely infection-naïve Hong Kong population, with around 70% of the population fully vaccinated[6]. We estimate a 3–14-fold relative decrease in detection rate at the beginning of wave five, even with high active surveillance, quarantine, and mandatory testing of building residents following case detection or discovery of virus in sewage. The implementation of local PHSMs initially reduced transmission rates, but the rate of infection increased at the start of the Chinese New Year public holiday and $R_e$ remained >1 for most of February 2022 causing unprecedented levels of local infection. According to our results, the February surge in reported cases was caused by numerous sustained transmission chains circulating prior to the Chinese New Year, rather than repeat introductions of BA.2.2 during this period. This shows that increased social mixing associated with holiday periods can increase the risk of resurgent outbreaks. Furthermore, the significant differences observed between $R_e$ and $R_t$ suggest under-ascertainment, co-circulation of multiple lineages, and/ or limited sequencing.

Increasing evidence shows superspreading plays a substantial role in SARS-CoV-2 transmission, with a small proportion of infected

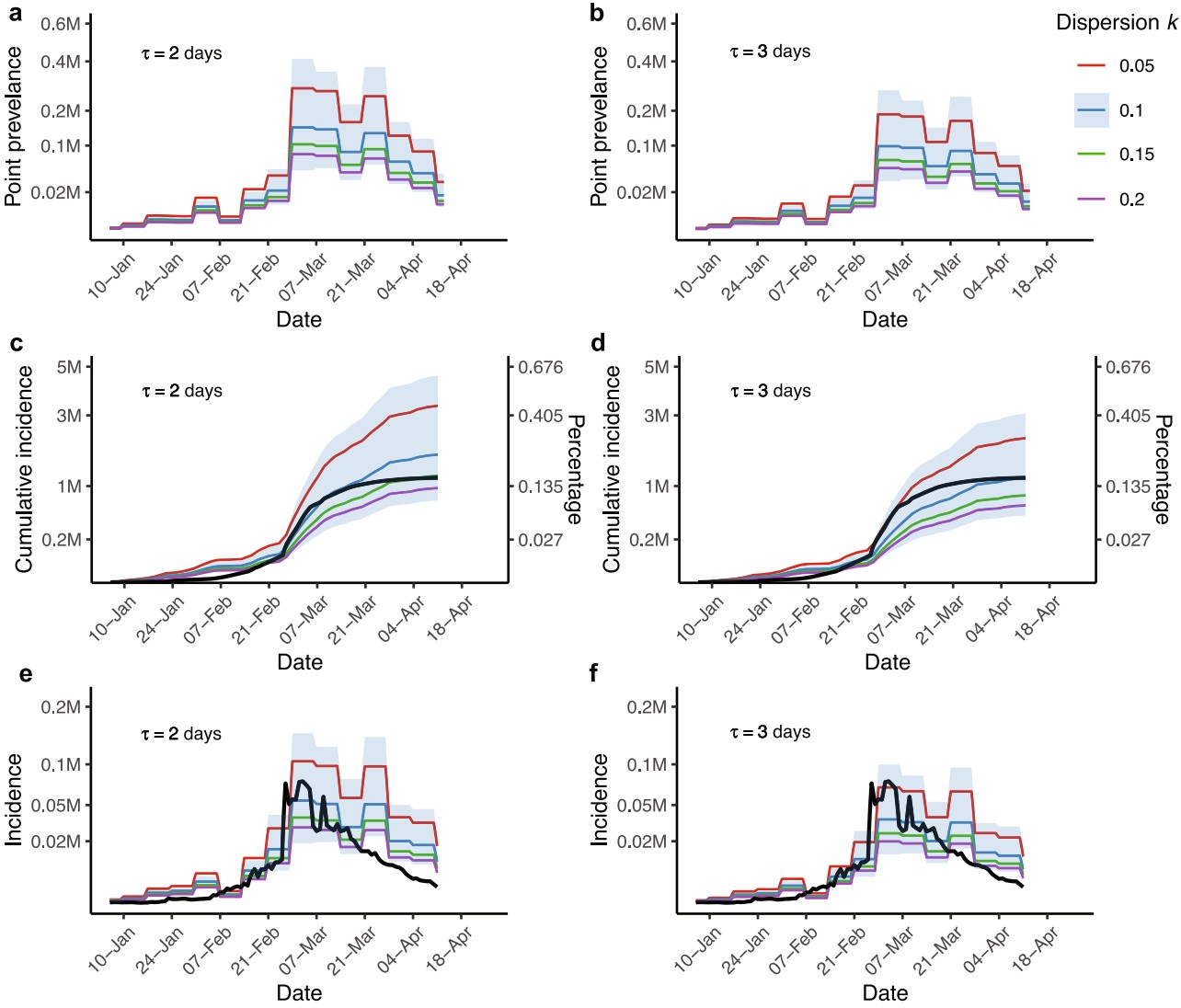

**Fig. 4 | Estimations of point prevalence and cumulative incidence from 6 January to 13 April.** Colored lines represent different levels of dispersion parameter $k$. Point prevalence, cumulative incidence, and incidence are shown at generation time $\tau = 2$ days in **a**, **c**, and **e**, and at $\tau = 3$ days in **b**, **d**, and **f**. The black lines in **c**–**f** are locally reported cases. Incidence and point prevalence were averaged weekly. The blue shaded area denotes the 95% confidence interval when $k = 0.1$. The scale for the incidence and percentage is the same in **c**, **d**.

individuals causing a large proportion of secondary cases. Previous studies estimated superspreading using the dispersion parameter $k$ in transmission clusters in the range of 0.06 to 2.97[22,23], while estimates using two clusters in Hong Kong between 2 and 21 January 2022 were around 0.2 and 0.33 for BA.1 and BA.2, respectively[24,25]. In more recent work, temporal changes in the dispersion parameter in Hong Kong was estimated to be closer to 0.1 when stringent PHSMs were in place[25]. Since most cases during January to April 2022 resulted from a single introduction of BA.2, we used phylodynamic models to compare the reported and estimated case numbers at varying degrees of over-dispersion. Our estimates of mean prevalence and cumulative incidence assuming the estimated dispersion of 0.1 at $\tau = 2$ days indicated a 49% underreporting rate, whereas a high dispersion ($k = 0.05$) showed a range of 89 to 184% underreporting. Interestingly, estimates of prevalence assuming extreme superspreading were similar to infections predicted by modeling efforts using case reporting, which predicted the Hong Kong epidemic trajectory with relative accuracy[26]. Alternatively, if we assume about 40% of Hong Kong's population (~3 million) contracted the virus during January–April 2022, a conservative estimate in comparison to real-time projections[26], we anticipate that superspreading will occur at coefficients below 0.1 indicating high overdispersion of cases. Furthermore, we observed the impact of COVID-19 was unequally felt across the 18 districts in Hong Kong. As such, specific measures should be considered to more effectively reduce morbidity and mortality: as high-density low-income areas were most impacted by COVID-19, while low-density, high-income areas were at greater risk of lineage introductions.

In the early stages of an outbreak, the reproduction number is commonly overestimated due to many factors[27], such as incorrectly accounting for imported cases and subpopulations with higher transmission rates. In this study, the first community case (Case B) was detected and imported cases were excluded via extensive contact tracing. Whether the intrinsic transmission rate of SARS-CoV-2 is higher in particular subpopulations (e.g. children and/or the elderly) in Hong Kong is unknown, and whether this could result in overly high estimates of reproduction numbers requires further study. In addition, our previous study[28], using comprehensive simulation analysis, showed our approach for $R_t$ estimation would tend to underestimate $R_t$ when $R_t$ is increasing, and overestimate $R_t$ when $R_t$ is decreasing, but could still provide the correct direction of change of $R_t$. In our study,

**Table 1 | Prevalence and incidence under various levels of generation time (τ) and dispersion (k)**

| Generation time τ (days) | Dispersion k | Total cumulative incidence (95% CI) | Underreporting rate[a] | Prevalence[b] (95% CI) | Incidence[b] (95% CI) |
|---|---|---|---|---|---|
| 2 | 0.05 | 3.35 (1.38, 7.40) | 183.90% | 0.28 (0.10, 0.79) | 0.10 (0.04, 0.29) |
| | 0.1 | 1.76 (0.72, 4.59) | 49.15% | 0.15 (0.05, 0.41) | 0.05 (0.02, 0.15) |
| | 0.15 | 1.22 (0.50, 3.20) | 3.39% | 0.10 (0.03, 0.29) | 0.04 (0.01, 0.10) |
| | 0.2 | 0.96 (0.39, 2.51) | - | 0.08 (0.03, 0.25) | 0.03 (0.01, 0.08) |
| 3 | 0.05 | 2.23 (0.92, 5.85) | 88.98% | 0.19 (0.06, 0.52) | 0.07 (0.02, 0.19) |
| | 0.1 | 1.17 (0.48, 3.06) | - | 0.10 (0.03, 0.27) | 0.04 (0.01, 0.10) |
| | 0.15 | 0.82 (0.34, 2.13) | - | 0.07 (0.02, 0.19) | 0.03 (0.00, 0.07) |
| | 0.2 | 0.64 (0.26, 1.67) | - | 0.05 (0.02, 0.15) | 0.02 (0.00, 0.05) |

[a]Calculated by the mean total cumulative incidence in comparison to 1.18 million reported cases.
[b]Million cases per day, at the inferred outbreak peak from 28 Feb to 6 March.

we have discussed how variable sampling of sequences throughout the outbreak could overestimate $R_e$ in the BDSKY model if unreliable prior assumptions of sampling proportions are used (Supplementary Note and Supplementary Figs. 9 and 10). These biases could account for the difference in $R_e$ and $R_t$ and have an impact on interpreting the dynamics of the fifth wave in Hong Kong.

Furthermore, GISAID sequence submission records between January and April 2022 show that sequencing in Hong Kong was typically completed within two weeks. However, the mean number of sequences submitted with a delay of less than 2 weeks was only 45 per week (median: 32; range: 1–194; Supplementary Fig. 8). This was inadequate considering the hundreds of confirmed daily case counts since February, when the total sampling proportion declined from ~30% to less than 1% (Supplementary Table 1). Underestimation of $R_e$ could occur if the sampling proportion is small, as observed since February, which failed to capture the entire genetic diversity revealed through $N_e$. When RAT-positive cases were included in public reporting from 26 February, a further sharp spike in $N_e$ followed. This suggests that BA2.2 sublineages that circulated cryptically were better captured. These observations indicate the timeliness and quantity of genomic surveillance in Hong Kong should be improved.

Overall, this study describes the origin, transmission dynamics, and impact of the largest SARS-CoV-2 wave in Hong Kong during a period of low population immunity and poor elderly vaccine uptake, providing a context for ongoing and future public health interventions. To help track epidemic dynamics and effectively manage the relaxation of PHSMs while accounting for the available capacity of the health system, it is necessary to enhance the genomic surveillance of SARS-CoV-2 in Hong Kong and develop a system that can evaluate and parameterize genomic and epidemiological data as close to real-time as possible. Ultimately, the effectiveness of PHSMs depends upon the ability to adapt to and respond to emerging and unpredictable health threats.

## Methods

### Genomic, epidemiologic, and human mobility datasets from Hong Kong

To elucidate the timing and origins of SARS-CoV-2 lineages during the fifth wave in Hong Kong, 116 saliva or nasopharyngeal samples from individual cases between 2 January and 4 February 2022, along with detailed epidemiological records including onset date, report date, and contact history were obtained from the Centre for Health Protection, Hong Kong. This study was conducted under ethical approval from the Institutional Review Board of the University of Hong Kong (UW 20–168). Because samples were collected as part of routine COVID-19 surveillance activities and were de-identified, a waiver of consent was granted. De-identified RT-PCR positive samples were sequenced using the same pipeline as in our recent studies[17,29]. Full-genome analysis was conducted at a World Health Organization reference laboratory at the University of

Hong Kong (Institutional Review Board no. UW 20–168). QIAamp Viral RNA Mini Kit (Qiagen, Cat. No.: 52906) was used to extract RNA. A number of gene-specific primers (https://github.com/Leo-Poon-Lab/mutations-under-sarscov2-vaccination/blob/main/Source%20Data/) targeting different regions of the viral genome were used to reverse transcribe the extracted RNA. For full-genome amplification, multiple overlapping 2-kb PCRs were performed with LA Taq DNA polymerase (Takara, Cat. No.: RR002M). The QIAquick PCR Purification Kit (Qiagen, Cat. No.: 28106) was used to purify PCR amplicons. DNA Prep (Illumina, Cat. No. 20018704) was used to prepare libraries from purified amplicons obtained from the same specimen. We quantified the libraries using Qubit dsDNA HS Assay Kits (Life Technologies, Cat. No.: Q32851) and sequenced them using Novaseq or iSeq100 sequencers (Illumina). All routine Hong Kong Delta and Omicron sequences deposited in GISAID until 30 April 2022 were also included. In addition, 10 random global (non-HK) sequences and 10 global sequences most similar by pairwise SNP distance to Hong Kong sequences per country per month from November 2021 to April 2022 were included (downloaded on 1 May 2022, Supplementary Data 4) as background to comprehensively and accurately define the monophyletic clade in Hong Kong and possible viral lineage exportations. Finally, reference genomes for each clade were included from GISAID (accessed on 8 May 2022, $n = 258$).

Pango lineage[30] was assigned to each sequence using Pangolin v.4.0.5, data version v1.3[31]. All nucleotide sequences were aligned to reference Wuhan-Hu-1 (GenBank accession MN908947.3), and those shorter than 27,000 nt were discarded. Duplicate sequences were removed, and sites deemed as problematic by other studies were masked (https://github.com/vjlab/omicronwave-hk) prior to phylogenetic analysis. Based on a regression of sample collection dates and root-to-tip genetic distances (from a maximum likelihood (ML) tree constructed in IQ-TREE 2[32] and rooted with Wuhan-Hu-1: GenBank accession MN908947.3), sequences that did not deviate more than eight interquartile ranges were considered as high quality and retained for subsequent analysis. As a result, 3317 Hong Kong sequences and 5220 international sequences were included.

Epidemiological trends of confirmed cases, PCR results, and control measures in Hong Kong between January to April 2022 (Fig. 2a) were obtained from Centre for Health Protection (https://www.chp.gov.hk/en/index.html). Given that over 90% of the daily journeys in Hong Kong are made using public transport[33], changes in mobility during January–April 2022 grouped by children, students, adults, and the elderly were obtained from Octopus cards, which are ubiquitously used by the Hong Kong population for daily public transport and small retail payments (https://www.octopus.com.hk/tc/consumer/index.html).

### Phylogenetic analysis
Bayesian time-scaled phylogenetic analyses were performed separately for Delta (HK = 126, global = 1426), Omicron BA.1.* (HK = 383,

global = 2234), and Omicron BA.2.* (HK = 2807, global = 1361), as they evolved from ancestral SARS-CoV-2 strains independently (Fig. 1a). Molecular clock rates used as priors for the full datasets were estimated from a subset of genomes sampled as evenly as possible across epidemiological weeks (Delta, $n = 150$; Omicron BA.1.*, $n = 181$; and Omicron BA.2.*, $n = 258$) using the HKY + G4 + I substitution model with a strict molecular clock model and an exponential coalescent tree prior for the Omicron lineages and a constant coalescent for Delta. Six independent Markov Chain Monte Carlo (MCMC) chains were each run for 100 million steps, discarding the first 10 million as burn-in and resampling states every 2000 steps.

Lineages resulting from independent introductions into the Hong Kong community were inferred by estimating monophyletic clades from the full datasets using a Bayesian molecular clock phylogenetic analysis pipeline[34] implemented in BEAST (v.1.10)[35] (commit:d1a45). ML trees with branches scaled to genetic distance in IQ-TREE 2[32] and time in TreeTime[36] were supplied as priors. Internal branches with less than one substitution were collapsed into polytomies. The analyses were run using a strict clock model with evolutionary rates estimated using the above subsampling datasets (Delta, $5.5 \times 10^{-4}$; Omicron BA.1.*, $3.79 \times 10^{-4}$; Omicron BA.2.*, $4.0 \times 10^{-4}$ substitutions/site/year), the Skygrid population model with weekly grid points and a Laplace root-height prior with mean equal to the time-calibrated tree estimated by TreeTime[36] was used, with scale set to 20% of the mean. For each analysis, we ran 40 MCMC chains of 40 million, sampling every 60,000 steps with the first 4 million discarded as burn-in. Model convergence of mixing chains was inspected in Tracer (v.1.7.1)[37] to ensure an effective sample size (ESS) of >200 for each parameter. Monophyletic clades in the posterior trees were identified using the R package "NELSI"[38]. It is notable that SARS-CoV-2 genomes with low variation among transmissions and our epidemiological data showed single introductions led to local outbreaks of HK-BA.2.2, HK-AY.127, and BA.1 (Dance cluster). Global sequences were therefore excluded when defining the three monophyletic clades. The R package "ggtree"[39] was used for tree visualization.

## Phylogeography of HK-BA.2.2

To infer migration patterns of HK-BA.2.2 in the global context, we used a two-state (HK and non-HK) asymmetric discrete-trait analysis model implemented in BEAST v.10.1.4 with a HKY + G4 + I substitution model, an uncorrelated relaxed molecular clock model (the prior of $4.0 \times 10^{-4}$ substitutions/site/year estimated for Omicron BA.2.*) with lognormal rate distribution (UCLN) and an exponential coalescent tree prior. For this analysis, we included the 10 earliest and 10 most recent sequences alongside 125 randomly selected cases from the HK-BA.2.2 monophyletic clade, 23 descendant sequences representing each country and province in mainland China, and two closely related ancestral BA.2.2 sequences (EPI_ISL_13330947 and EPI_ISL_9897214). We removed further outliers using TempEst v.1.5.3[40] under the premise that there is no major difference between the time signal of the dataset before and after sampling. As contact tracing and confirmatory phylogenetic analysis showed that HK-BA.2.2 virus was first detected in an international traveler arriving on 4 January 2022, an informative Laplace tMRCA of HK-BA.2.2 monophyletic clade prior with a mean (M) of 0.312 and a variance (s) of 0.01 was chosen. Six independent MCMC chains with 40 million states were performed, sampling every 2000 and discarding 10% as burn-in. As a result, 108,000 time-calibrated posterior trees were generated and used as an empirical distribution for the phylogeographic analysis. We combined two independent chains, each run for five million MCMC steps, sampling 1000 steps and discarding 10% as burn-in.

## Effective population size ($N_e$) and relative case detection rate

For the largest monophyletic clade (HK-BA.2.2, $n = 2455$) in Hong Kong, the above Bayesian molecular clock phylogenetic analysis

pipeline with a strict clock fixed to $5.5 \times 10^{-4}$ substitutions/site/year (mean value estimated from relaxed clock rate in phylogeography of HK-BA.2.2) was repeated to estimate changes in the effective population size ($N_e$) using Skygrid population model. Following Smith et al.[41], by combining $N_e$ and the epidemiological information of conducted tests, we can estimate the dynamics of the relative case detection rate:

$$P_t(\text{tested|infected}) = \frac{P_t(\text{tested|infected}) * P_t(\text{tested})}{P_t(\text{infected})} \quad (1)$$

subject to

$$P_t(\text{infected|tested}) = \frac{r_{\text{pos}} - (1 - \text{spec})}{\text{sens} - (1 - \text{spec})} \quad (2)$$

$$P_t(\text{infected}) = \frac{\text{pop}_{\text{infected}}}{\text{pop}} = \frac{c * N_e}{\text{pop}} \quad (3)$$

$$P_t(\text{tested}) = \left(1 - \left(1 - \frac{1}{\text{pop}}\right)^{n_t}\right) \quad (4)$$

where $\text{pop}_{\text{infected}}$ is the number of infections in the population which can be simplified as a constant factor ($c$, which represents the number of true cases per effective population 'unit') times $N_e$ due to their linear correlation. pop is population size (7.4 million) in Hong Kong, $r_{\text{pos}}$ denotes the positivity rate of the PCR tests conducted, and $n_t$ represents the number of tests conducted. Sensitivity *sens* and specificity *spec* were set to 1 as the reported COVID-19 cases until 26 February were confirmed twice by PCR tests. However, reducing *sens* does not change the dynamics of the relative case detection rate, but has an overall increase in the $y$ axis in Fig. 3c.

## Effective reproduction number ($R_e$)

For improved computational efficiency and tested the effect of sub-sampling schemes (Supplementary Note and Supplementary Fig. 9) in constructing $R_e$, we used the sampling schemes recommended by the WHO for practical use in different settings and scenarios[42,43], which included uniform and proportional sampling, to construct three datasets ($n = 262$, uniform: 20 sequences per week; $n = 502$, uniform: 40 sequences per week; $n = 897$, proportional) summarized in Supplementary Fig. 11. A birth-death skyline serial (BDSS) model[18] implemented in BEAST (v.2.6.7)[44] was used to infer the dynamics of the effective reproduction number ($R_e$). The HKY + G4 substitution model and a strict clock fixed to $5.5 \times 10^{-4}$ substitutions/site/year (mean value estimated from relaxed clock rate in phylogeography of HK-BA.2.2) were used. Given that the BDSS model is affected by biases from sampling proportion (as shown in the sensitivity analysis in the Supplementary Note) and uneven sampling during the sequencing period from January to April, we assume that $R_e$ and the sampling proportion are piecewise constant functions over 16 time intervals, roughly corresponding to weeks between 3 January and 26 April. Specifically, we assume that the sampling proportion per week is 0 before the collection time of the oldest sample, and is given a uniform distribution as prior with an upper bound on the empirical ratio of the number of subsampling sequences per week to the number of weekly reported cases. However, due to extensive sequencing done during the second week from 10 to 17 January (Fig. 1), where very few BA.2.2 cases were reported, the lower bound of sampling proportion prior was set at 0.3 (Supplementary Table 1, the upper and lower bounds of the sampling proportion prior could lead to a higher $R_e$ between 10 and 17 January). A non-informative prior for *t*Origin with lower bound set to 1 December 2021 was chosen. A lognormal prior with a mean of 0.0 and a variance (S) of 1.0 was set for $R_e$. To test the effect of the prior on $R_e$, we compared different levels of variance S (2 and 3) and found no

significant differences to $R_e$ shown in Fig. 3. Given that individuals who test positive in Hong Kong will be isolated, we assumed that there will be no further transmission from these individuals in our analysis. If this assumption is not valid, it could lead to an overestimation of the death rate and consequently the underestimation of $R_e$. The MCMC runs were performed for at least two independent chains of 100–200 million generations, sampling every 10,000 steps, with at least 10% discarded as burn-in. The R package "bdskytools" (https://github.com/laduplessis/bdskytools) was used to plot changes in $R_e$ over time. The final $R_e$ was selected from the estimation using the uniform subsampling dataset (40 sequences per week), which was better matched with the trend of $R_t$ (Supplementary Fig. 10).

### Instantaneous effective reproduction number ($R_t$)
We computed $R_t$ based on local cases and those epidemiologically linked to local cases, as defined by the Centre for Health Protection (CHP, https://www.coronavirus.gov.hk/eng/index.html). As SARS-CoV-2 can transmit pre-symptomatically[45], reconstructing incidence by date of infection provides a more accurate estimate of $R_t$[46]. Therefore, we reconstructed the epidemic curve by infection date based on confirmation date with the distribution of delay from infection to confirmation using a deconvolution approach[28]. We conducted the inference in a Bayesian framework and developed Markov chain Monte Carlo algorithms to estimate the posterior distribution of the model parameters and used a bootstrap approach to account for uncertainty associated with deconvolution[47]. As Cori et al.[46] and Parag et al.[48] show, $R_t$ measures the average transmissibility over a time window of length $\tau$ ending at time $t$ under the assumption that $R_t$ is constant within this time window, where $\tau$ is the smoothing parameter. In this study, we take $\tau = 14$, to avoid unstable estimates for time-varying reproduction number. Correspondingly, the estimated $R_t$ would need a few days to move to its true value, but still provide the correct direction of change[28].

### Estimation of prevalence and incidence
Given the complex dynamics of the fifth wave in Hong Kong, we estimated point prevalence ($I$) from $N_e \tau$, following a discrete generation model with arbitrary offspring distribution and changing population size[49]. Due to the superspreading dynamics at SARS-CoV-2[21,50], a negative binomial offspring distribution was assumed, for which dispersion parameter ($k$) controls its shape. Point prevalence ($I$) can be calculated using the formula below:

$$I = \frac{N_e}{\tau} * (\sigma^2/R + R - 1) \tag{5}$$

subject to:

$$\sigma^2 = R + R^2/k \tag{6}$$

where $N_e$ is the effective population size scaled by the generation length per year[51] corresponding to $N_e \tau$, where $\tau$ denotes generation time, $R$ is the mean number of secondary cases, and $k$ is the dispersion parameter of secondary cases. In this study, we used $N_e$ estimated by a Skygrid coalescent model with 95% confidence interval (CI), $\tau = 2$ and 3 days[4], $R$ replaced by the estimation of mean $R_e$ using a BDSKY model and $k$ from 0.05 to 0.2 (median = 0.1)[23]. Cumulative incidence was calculated by adding the prevalence of each serial interval (2.72 days[4]) together, with the 95% CI restricted to the total population size (7.4 million). Daily incidence was calculated by diminishing cumulative incidence.

### Reporting summary
Further information on research design is available in the Nature Portfolio Reporting Summary linked to this article.

## Data availability
Hong Kong SARS-CoV-2 genome sequences and associated metadata generated in this study are deposited at GenBank and GISAID (accession numbers are available on GitHub at https://github.com/vjlab/omicronwave-hk/blob/v.1.0.0/data/). The aggregate data of passenger numbers by public transportation means were provided by Octopus Cards Limited (Octopus). We obtained consent from Octopus to share the aggregate data of transport transactions between 1 January and 30 April 2022. Our agreement with Octopus prohibits us from further sharing data with third parties, but interested parties may contact Octopus.

## Code availability
All anonymized data, code, and analysis files are available on GitHub: https://doi.org/10.5281/zenodo.7804170.

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

## Acknowledgements

We acknowledge the technical support provided by colleagues from the Centre for PanorOmic Sciences of the University of Hong Kong. We also acknowledge the Centre for Health Protection of the Department of Health for providing epidemiological data for the study. The computations were performed using research computing facilities offered by Information Technology Services, at the University of Hong Kong. We gratefully acknowledge the staff from the originating laboratories responsible for obtaining the specimens and from the submitting laboratories where the genome data were generated and shared via GISAID (Supplementary Data 4). We thank Octopus Cards Limited for providing aggregate data of passenger numbers by public transportation means for the research. The funding bodies had no role in the design of the study and collection, analysis, and interpretation of data and writing of the manuscript. Funding: National Institutes of Health contract number 75N93021C00016 (V.D., L.L.M.P.), Research Grants Council of the Hong Kong SAR, China (Project no. [T11-705/21-N]) (L.L.M.P.), Health and Medical Research Fund, Food and Health Bureau of the Hong Kong SAR Government (COVID190205) (L.L.M.P.).

## Author contributions

V.D. conceived and designed the research. R.X., K.M.E. curated the Hong Kong epidemiological case data. D.Y.M.N, G.Y.Z.L., P.K., L.D.J.C., S.M.S.C., performed sample characterization, and genome sequencing. G.K.H.S., M.P., L.L.M.P. supervised sample characterization and genome sequencing. R.X., S.G., X.W., and H.G. designed and implemented

genomic data processing pipelines. R.X. performed a phylodynamic analysis. D.C.A., B.J.C., V.D. advised on genomic epidemiology. K.S.M.L. summarized human mobility data. T.K.T., W.X. performed real-time epidemiologic modeling. J.T.W., G.M.L., B.J.C. supervised real-time epidemiologic modeling. R.X., K.M.E., and V.D. wrote the first draft of the manuscript. All authors read and approved the manuscript.

## Competing interests

The authors declare no competing interests.
