## [Peer Review File · Nature Communications]

Resurgence of Omicron BA.2 in SARS-CoV-2 infection-naive Hong KongREVIEWER COMMENTS

Reviewer #1 (Remarks to the Author):

In this work, Xie et al. performed an extensive analyses of the origin and transmission dynamics of the fifth COVID-19 wave in Hong Kong between Jan and Apr 2022. The key thesis of this work is that effective reproductive rate (R_e) estimated from phylodynamic analyses of BA.2.2 in Hong Kong was far greater than instantaneous effective reproductive rate (R_t) estimated from case ascertainment between 9 and 19 January 2022. They then estimated the effective population size (N_e) to compute that relative case detection rate likely decreased by 3-14 fold between 15 Jan and 4 Feb 2022 and that underreporting was likely only alleviated after the incorporation of diagnoses from rapid antigen tests from 26 Feb 2022. They then estimated prevalence based on their effective reproduction number analyses and determined that there was likely substantial underreporting of cases and that extensive superspreading occurred during the fifth COVID-19 wave in Hong Kong.

Overall, while the manuscript is nicely written with excellent figures, I have several major concerns if the presented evidence support the authors' conclusion that the reliance on R_t estimation methods can be misleading at times (which I actually do agree with).

Major

1. Given how disparate/uneven sampling for sequencing was over time (Fig 1), the authors should be more careful with characterizations like "152 detections did not lead to onward transmission". Unless extensive contact tracing and sequencing are performed, I am not sure the authors can be confident if the sampled individuals did not go on to transmit to others. Similarly, the authors cannot conclude based on their analyses that the exported viral lineage did not go on to propagate extensively in other countries.

2. In the same vein, statements like "We also found that the HK-BA.2.2 lineage was exported from Hong Kong to at least nine other countries but did not become widespread elsewhere" are rather careless. The authors only included non-Hong Kong sequences in their analyses by only randomly subsampling 20 global sequences per country per month. While this is fine as a comparison of the genetic diversity found in Hong Kong against what was sampled globally, the authors cannot make further inferences using their analyses about how much further the viruses exported from Hong Kong propagated elsewhere.

3. Given that there were other substantial NPIs (i.e. entertainment venues closed on 6 Jan and face-to-face teaching suspended for kindergarten and primary school on 14 Jan) during early/mid-January, I am surprised by the extraordinarily high R_e estimate (9.5 and up to 14.0 according to HPD) during this time. The sharp drop in estimated R_e to 1.0 by 21 Jan is also quite jarring and the authors seemed to attribute this to the suspension of face-to-face teaching of secondary school children, leading to lower mobility levels among students (computed using public transport data). First of all, public transport is only one aspect of mobility and I wonder how much can we extrapolate from this data. More information on how R_e was estimated is also needed: specifically what were the BDSKY prior distributions used for sampling proportions? The BDSKY serial model, by default, assumes that a lineage is removed from the tree upon sampling which is unlikely the case for SARS-CoV-2 spread (i.e. there could still be transmissions after patients were tested/sampled). Was this assumption relaxed?

4. To that end, I found R_t to actually be a more realistic reflection on the timeliness of impact from NPIs that were introduced/relaxed. Instead of falling sharply, R_t gradually decreased when face-to-face teaching of both primary and secondary school children was suspended and increased gradually upon the impending and on Spring Festival holidays.

5. I also found the evidence presented for under-ascertainment of cases conflicting and confusing. Based on the R_e estimates and comparisons to R_t , the authors suggest that there were weeks of case underreporting during this mid-January. However, incidence estimates (Fig 4c-f), given different dispersion factors, were largely in line with reported case numbers. Case underreporting based on Figs 4e-f was only substantial from March onwards.

6. On that note, the formula used by Bedford estimate prevalence/incidence from N_e is actually from Koelle and Rasmussen (2012; <https://www.ncbi.nlm.nih.gov/pmc/articles/PMC3306638/>). I am unsure if this is the right approach to compute incidence from N_e because this was formulated under assumption that disease dynamics were at their endemic equilibrium, which is certainly not the case here under exponential growth).

Minor

7. The authors claimed that "at least four to five instances of hamster-to-human transmission were observed according to the Bayesian maximum clade credibility tree and the maximum likelihood phylogeny, respectively (Supp fig.2)." While there have been likely hamsters-to-human transmission events reported previously, I am unsure if the trees presented by authors suggest that. At best, it shows that there is a shared common ancestor between the sampled viruses in humans and hamsters but the analyses neither show the directionality of transmissions nor the likely host species of the common ancestor. I actually find this point to be rather distracting as I read through the paper and would advise the authors to take this part out.

Reviewer #2 (Remarks to the Author):

The approach of this paper – to examine and track underlying drivers of transmission behind COVID-19 variants (particularly omicron sub-variants) in Hong Kong, which presents a unique mix of age-stratified susceptibility given its earlier measures, is interesting and of potentially wider importance. However, I have major concerns about the methodology, interpretation and validation of the results and in particular the conclusions drawn from the estimates of R_e and R_t . These issues need to be resolved for this work to be convincing, especially as the central claims in this paper follow from the analyses in question.

Specifically, not enough is done to confirm the accuracy of the R_e of 9.5, which appears to be the basis of the key conclusion of the paper about under-reporting. There are several issues:

1. The birth-death skyline is well known to suffer biases from sampling proportions (as all the parameters in this model are not identifiable). While checks against the prior on R_e are important and provided – similar examination of the sampling proportion prior choice and sensitivity tests are needed. This is especially important given that it can have a large impact on R_e .

2. While R_e is drawn from BA.2 sequences it is compared to an R_t inferred from all local cases (and if it is not then results concerning potential co-circulating lineages later in the time series seem odd). The large R_e occurs around Jan 10-17 but there are also few sequences and cases of BA.2 in this period. This is when reproduction numbers are most difficult to infer, especially resurgences. Consequently, validation of these estimates would help e.g., by showing how well R_e or R_t can sequentially reproduce its data e.g.,

<https://www.sciencedirect.com/science/article/pii/S0022249606000058> and

<https://sites.stat.washington.edu/raftery/Research/PDF/Gneiting2007jrssb.pdf>

3. There are many potential explanations for a large initial R_e that could also explain this result e.g., <https://onlinelibrary.wiley.com/doi/10.1002/sim.4174> but not all of them are ruled out or discussed. Strengthening the rejection of these other hypotheses would be useful if the claim that the difference in R_e and R_t is largely due to sampling.

4. However, even if these confounders are not important the claim itself only works if a significant change in reporting rates occurs. There is such a change from PCR to RAT but this occurs much later. As <https://academic.oup.com/aje/article/178/9/1505/89262> and <https://www.nature.com/articles/s43588-022-00313-1> show, much of the information about R_t is preserved in periods of relatively constant rate. These points need to be discussed and qualified especially with regards to the timing of effects.

5. The rise in N_e occurs in line with the major fall in R_e – how can this be explained? If N_e is proportional to BA.2 infections and its growth rate increases, then one would not expect R_e to fall to 1 or below at that time (even if generation times change).

6. It also appears that the imposition of the school face-to-face intervention correlated with a fall in R_t but a fall in R_e had already occurred notably earlier. This would also need to be explained

given that timing of events is often included in the results.

7. The sampling of sequences in constructing R_e is also important – what schemes were used and were any validation tests across different sequence undersampling schemes considered? This choice can strongly influence R_e .

8. Are there differences in the definitions of R_e and R_t ? The instantaneous vs effective terminology is used but not explained

Reviewer #3 (Remarks to the Author):

The paper describes the epidemiology of a large wave of predominantly Omicron BA.2 cases, in a relatively immunologically naïve population. This resulted in high mortality, despite implementation of multiple non-pharmaceutical interventions. The results show the importance of keeping track of unbiased case rates and effective reproduction numbers in near-real time, to avoid ill-informed policy decisions. A number of important lessons can be learned from this study, including the need for timely sequencing of confirmed cases and subsequent analysis of genomic and epidemiologic data - it's clear that data were not available (or analysed) in time for crucial public health decisions to be made.

The methods seem generally sound based on my understanding of the field, although I do have a number of questions and comments:

When were the samples sequenced and made available for analysis relative to the outbreak? What explains the highly variable proportion of notified cases sequenced throughout the outbreak, and why were so few sequenced from mid-February onwards? It would be helpful to understand more about the application of genomic surveillance in Hong Kong and how that changed over time, and discuss the implications of such highly variable sampling of sequences on estimates of R_e and N_e (i.e. could it have biased estimates?).

In Figure 1 I found the situation concerning RATs confusing - there is a marker on March 7th that says when the online platform for reporting RATs was introduced but there are a large number of RATs reported after Feb 26th when the confirmed cases included RATs from private hospitals and laboratories. Did the system change to include RATs from sources other than private hospitals and laboratories on March the 7th? Some clarity in the figure and/or figure legend would make the situation clearer concerning RATs. Also I suggest you say that the spike on March the 9th includes cases from the 8th to make it clearer that this is an artifact in the data.

The effective reproductive rate (R_e) and the instantaneous effective reproduction rate (R_t) could be more clearly defined earlier in the text, and described more clearly in the methods. Note in epidemiological terms they are numbers and not true rates and should really be named as such. If I understand it correctly, R_e is determined from the birth-death model using sampled sequence data and (presumably), sample dates, whereas R_t is estimated from reported incidence data, after imputing infection dates.

I recommend defining R_e and R_t briefly in the results section, as well as in more detail in the methods section. For example in the results section they could be defined as R_e = effective reproduction number estimated using sampled sequences and sample dates over XX time intervals (i.e. indicate what time grid was used in bdskytools), and R_t = instantaneous effective reproduction number calculated daily using the estimated number of infections on each day.

Page 17: The last two equations seem to be mislabelled. Shouldn't $P(\text{tested})$ be $1-(1-1/\text{pop})^{\text{nt}}$ and $P(\text{infected})$ be $c*N_e/\text{pop}$?

Assuming this is correct, how is c defined in biological terms? Presumably it represents the number of true cases per effective population 'unit'?

Minor comments:

Page 3: "peak of 3.5 per 100,000 people" please give units (per day?)

Page 6: "...lineage introduced by from imported..." should be "lineage introduced by imported..."

Page 6: "Delta cases detected in the community between 15 January and late March" I can't see any evidence of Delta cases in March in Figure 1 or Supp Figure 2?

Page 6: "152 detections did not lead to onward transmission" suggest "152 detections did not lead to detectable onward transmission". Important given the low proportion of cases sequenced.

Page 8: "...of viruses with spike I1221T viruses was..." repetition of "viruses"

Page 8: "...as indicated by the light blue dashed line" should be red dashed line.

Figure 3a - y axis should be labelled R_e/R_t and it should be made clear in the text and the legend that R_t refers to all cases, not just BA2.2 (assuming that is the case).

Page 13: "...indicated a 15–73% under detection, whereas a high dispersion ($k = 0.05$) showed a 2–4-fold under detection of cases". Suggest you provide underreporting rates in the text, as in Table 1, or relative underreporting rates i.e. be consistent in either % under detection or X fold relative under detection.

Responses to Reviewer Comments

General response to reviewers' comments.

We are grateful to the reviewers for highlighting the significance of our study and for the constructive feedback, particularly in R_e estimation and interpretation of the differences between R_e and R_t . We have addressed each of the comments below and made corresponding changes in the revised manuscript.

Reviewer #1 (Remarks to the Author):

In this work, Xie et al. performed an extensive analyses of the origin and transmission dynamics of the fifth COVID-19 wave in Hong Kong between Jan and Apr 2022. The key thesis of this work is that effective reproductive rate (R_e) estimated from phylodynamic analyses of BA.2.2 in Hong Kong was far greater than instantaneous effective reproductive rate (R_t) estimated from case ascertainment between 9 and 19 January 2022. They then estimated the effective population size (N_e) to compute that relative case detection rate likely decreased by 3-14 fold between 15 Jan and 4 Feb 2022 and that underreporting was likely only alleviated after the incorporation of diagnoses from rapid antigen tests from 26 Feb 2022. They then estimated prevalence based on their effective reproduction number analyses and determined that there was likely substantial underreporting of cases and that extensive superspreading occurred during the fifth COVID-19 wave in Hong Kong.

Overall, while the manuscript is nicely written with excellent figures, I have several major concerns if the presented evidence support the authors' conclusion that the reliance on R_t estimation methods can be misleading at times (which I actually do agree with).

Response to Reviewer 1's general remarks We appreciate the compliments on the writing and figures. In response to the concern raised about the conclusion that reliance on R_t estimation methods can be misleading, we have conducted additional sensitivity tests on our R_e estimates. The results of these tests have been added to the revised manuscript in the form of a supplementary note and figure. Our tests revealed that previously defaulted priors for sampling proportion could lead to an overestimation of R_e due to convergent high sampling proportions. Therefore, we have adjusted these priors in order to find greater similarity between R_e and R_t . However, we found periods with significant differences between R_e and R_t , which could be due to factors such as under-reporting, co-circulation of multiple lineages, and/or limited sequencing. We have provided more details on this in our response to Comment 3 below.

Reviewer 1, comment 1. Given how disparate/uneven sampling for sequencing was over time (Fig 1), the authors should be more careful with characterizations like "152 detections did not lead to onward transmission". Unless extensive contact tracing and sequencing are performed, I am not sure the authors can be confident if the sampled individuals did not go on to transmit to others. Similarly, the authors cannot conclude based on their analyses that the exported viral lineage did not go on to propagate extensively in other countries.

Response to R1C1 We acknowledge Hong Kong was unable to perform extensive contact tracing and sequencing, especially since February, when unprecedented case numbers were reported. Correspondingly, we replaced "onward transmission" to

“detectable onward transmission” (lines 94 and 106) as suggested by reviewer 3 (minor comment 4).

Reviewer 1, comment 2. In the same vein, statements like “We also found that the HK-BA.2.2 lineage was exported from Hong Kong to at least nine other countries but did not become widespread elsewhere based on a ” are rather careless. The authors only included non-Hong Kong sequences in their analyses by only randomly subsampling 20 global sequences per country per month. While this is fine as a comparison of the genetic diversity found in Hong Kong against what was sampled globally, the authors cannot make further inferences using their analyses about how much further the viruses exported from Hong Kong propagated elsewhere.

Response to R1C2. We can confirm that this lineage (HK BA.2.2) was not successful in other countries it has been detected in so far, as our analysis was comprehensive. For phylogenetics, we included ten randomly selected sequences and ten most similar sequences to each Hong Kong sequence from November 2021 to April 2022, per country per month. We also compared the predominance of the HK-BA.2.2 in newly seeded regions by comparing it with other variants reported during the subsequent period. Our results were cross-checked using Taxonium (<https://cov2tree.org/>) [1], an online resource that maintains the world’s largest SARS-CoV-2 phylogenetic tree with over 6 million sequences collected globally.

Based on this analysis, we found that in eight of nine countries where HK-BA.2.2 was detected, the proportion of HK-BA.2.2 to all SARS-CoV-2 sequences was less than 0.5%, indicating limited local transmission. The exception was mainland China, where the available sequence data was not sufficient to draw conclusions. To clarify these points, we revised lines 135-138 in Results, lines 331-335 in Methods, and Figure 2 legend as follows:

“We also found that the HK-BA.2.2 lineage was exported from Hong Kong to at least nine other countries but did not become widespread elsewhere as indicated by a very low proportion (less than 0.5%) of BA.2.2 sequences relative to all sequences, except in mainland China (25/83, ~30%) where insufficient sequences were available”

“The numbers in the brackets indicate BA.2.2 sequences for each country over the total number of sequences in GISAID (accessed 1 May 2022) from January to April 2022.”

“In addition, 10 random global (non-HK) sequences and 10 global sequences most similar by pairwise SNP distance to Hong Kong sequences per country per month from November 2021 to April 2022 were included (downloaded on 1 May 2022, Supplementary Data 4) as background to comprehensively and accurately define the monophyletic clade in Hong Kong and possible viral lineage exportations.”

Reviewer 1, comment 3. Given that there were other substantial NPIs (i.e. entertainment venues closed on 6 Jan and face-to-face teaching suspended for kindergarten and primary school on 14 Jan) during early/mid-January, I am surprised by the extraordinarily high Re

estimate (9.5 and up to 14.0 according to HPD) during this time. The sharp drop in estimated R_e to 1.0 by 21 Jan is also quite jarring and the authors seemed to attribute this to the suspension of face-to-face teaching of secondary school children, leading to lower mobility levels among students (computed using public transport data). First of all, public transport is only one aspect of mobility and I wonder how much can we extrapolate from this data. More information on how R_e was estimated is also needed: specifically what were the BDSKY prior distributions used for sampling proportions? The BDSKY serial model, by default, assumes that a lineage is removed from the tree upon sampling which is unlikely the case for SARS-CoV-2 spread (i.e. there could still be transmissions after patients were tested/sampled). Was this assumption relaxed?

Response to R1C3. We thank the reviewer for these observations and comments on R_e estimates and its attribution to the suspension of face-to-face teaching. In response, we have conducted sensitivity tests for the various priors to provide a more robust estimation of R_e , and made corresponding revisions in the manuscript (detailed below).

Sensitivity tests for different sampling proportion priors found that the default priors could result in an overestimation of R_e due to convergent high sampling proportions as noted in Supplementary Note and Supplementary Fig. 9. To address this, we adjusted the priors for our R_e estimation assuming that R_e and the sampling proportion are piecewise constant functions over 16-time intervals, which correspond to weeks between 3 January and 26 April. Further, we gave the sampling proportion a uniform prior distribution, with the upper bound being the empirical ratio of the number of subsampling sequences per week to the number of weekly reported cases, and the lower bound being set at 0.3 due to extensive sequencing done during the second week from 10 to 17 January when very few BA.2.2 cases were reported. The revised prior distributions are detailed in the Methods section of the manuscript.

Methods lines 433-443 “Given that the BDSS model is affected by biases from sampling proportion (as shown in the sensitivity analysis in the Supplementary Note) and uneven sampling during the sequencing period from January to April, we assume that R_e and the sampling proportion are piecewise constant functions over 16 time intervals, roughly corresponding to weeks between 3 January and 26 April. Specifically, we assume that the sampling proportion per week is 0 before the collection time of the oldest sample, and is given a uniform distribution as prior with an upper bound on the empirical ratio of the number of subsampling sequences per week to the number of weekly reported cases. However, due to extensive sequencing done during the second week from 10 to 17 January, where very few BA.2.2 cases were reported, the lower bound of sampling proportion prior was set at 0.3 (Supplementary Table 1).”

With the optimized priors, R_e showed a similar dynamic pattern to R_t , both confirming that the transmission rate was contained after the suspension of face-to-face teaching for secondary school children, which led to reduced mobility levels among students. It is worth noting that the timing of changes in R_e and R_t may be differ, while R_e changing weekly and R_t changing daily in our analysis. The significant differences observed between R_e and R_t at some time points could be explained by under-reporting, co-circulation of multiple lineages, and/or limited sequencing. These are summarized in the Results as follows.

Results, lines 154-184: “To reveal changes in the spread of BA.2.2 in Hong Kong over time, we used a Bayesian birth-death skyline model that explicitly estimates the rate of transmission, recovery, and sampling, enabling a direct inference of the effective reproductive number based on sampled sequences and sample dates over 16 time intervals, roughly corresponding to weeks between 3 January and 26 April (Fig. 3a). We observed an increase in the effective reproductive number (R_e) to 2.5

(HPD, 1.1–4.2) during the second week (10–16 January 2022), briefly matching the time point on 10 January when Case B left the hotel and introduced the virus into the community. Higher values of R_e (mean, 3.4; HPD, 2.2–4.8) continued to be observed until around 24 January, during the third week. The instantaneous effective reproduction number (R_t), estimated from the number of local infections reported per day, increased gradually from 1 (HPD, 0.6–2.2) on 12 January and peaked at 5.2 (HPD, 3.9–7.7) on 20 January. Although R_t was able to reproduce the dynamics of reported cases better than R_e during 10–24 January 2022 (Supplementary Note and Supplementary Fig. 4), the increase in R_e , as well as the potential for uncaptured transmission at the hotel, suggest that some community transmission chains were not identified. However, during the third week (17–23 January 2022), R_e was lower than R_t , which is most likely due to the co-circulation of multiple lineages (AY.127, BA.1* and BA.2*) (**Fig. 1**).

R_e decreased to 0.8 (HPD, 0.04–1.9) during the fourth week from 24–30 January 2022, consistent with the suspension of face-to-face teaching for kindergarten and primary schools by 14 January and secondary schools by 22 January, which substantially reduced mobility levels among students in Hong Kong (Supplementary Fig. 5). However, R_e increased again during the fifth week (31 January to 6 February 2022) to 2.7 (HPD, 1.8–3.7) in correlation with a slight increase in mobility levels during the Spring Festival holidays (1–3 February 2022). There was a similar dynamic pattern in R_t , but from 7 to 28 February, R_t remained above 2 with a slight decrease, significantly higher than the R_e that fluctuated around 1. Higher R_t may reflect the inclusion of cases caused by other co-circulating lineages (e.g. BA.2.10.1) (**Fig. 2c**) in which R_e has not been considered, or it may reflect under-sequencing of HK-B.A.2.2 during this period. Especially after 14 February, when infections overwhelmed the health care system, isolation facilities, and track-and-trace capacity, less than 1% of samples were sequenced (**Figs. 1 and 3a** and Table S1). Interestingly, from March to mid-April, R_e continued to fluctuate around 1 in comparison to R_t , which was less than 1, indicating a slower decline of the outbreak than anticipated.”

In terms of the utility of Octopus cards in detecting changes in mobility, Octopus cards are used ubiquitously for public transportation and small retail payments in Hong Kong [2] (<https://www.octopus.com.hk/tc/consumer/index.html>). Over 90% of daily journeys are made using the highly developed and sophisticated public transport in Hong Kong, the highest rate in the world [3]. These are clarified in Lines 349-353.

“Given that over 90% of the daily journeys in Hong Kong are made using public transport, changes in mobility during January–April 2022 grouped by children, students, adults, and the elderly, were obtained from Octopus cards, which are ubiquitously used by the Hong Kong population for daily public transport and small retail payments (<https://www.octopus.com.hk/tc/consumer/index.html>).”

Finally, while transmission after sampling is common, further transmission from detected cases is low in Hong Kong because individuals who test positive are isolated. However, these assumptions cannot be used directly in BEAST [4]. If we assume that the probability of becoming non-infectious after transmission <1 , sampled individuals may occur along branches, whereas in BEAST, the sampled individuals must be the tips of the tree. As a result, the death rate will be overestimated, leading to an underestimation of R_e . We clarified this assumption and discussed the limitations in lines 446-449:

“Given that individuals who test positive in Hong Kong will be isolated, we assumed that there will be no further transmission from these individuals in our analysis. If this

assumption is not valid, it could lead to an overestimation of the death rate and consequently the underestimation of R_e .”

Reviewer 1, comment 4. To that end, I found R_t to actually be a more realistic reflection on the timeliness of impact from NPIs that were introduced/relaxed. Instead of falling sharply, R_t gradually decreased when face-to-face teaching of both primary and secondary school children was suspended and increased gradually upon the impending and on Spring Festival holidays.

Response to R1C4. We agree that R_t provides daily updates, making it more accurate observation of the timeliness of impact of NPIs. However, as highlighted in our Response to Comment 3 above, R_e increased earlier showing potential for R_t estimation methods that are dependent on case reporting to miss key dynamic information during the outbreak. A combination of R_t and R_e , inferred from both epidemiological and genomic data, provides the most comprehensive inference of the dynamics of the fifth wave of COVID-19 in Hong Kong.

Reviewer 1, comment 5. I also found the evidence presented for under-ascertainment of cases conflicting and confusing. Based on the R_e estimates and comparisons to R_t , the authors suggest that there were weeks of case underreporting during this mid-January. However, incidence estimates (Fig 4c-f), given different dispersion factors, were largely in line with reported case numbers. Case underreporting based on Figs 4e-f was only substantial from March onwards.

Response to R1C5. To address this, we have improved the evidence presented for under-ascertainment by modifying the relevant sections (lines 182-184, 192-195, 220-224, below). According to the updated results, we show that from March to mid-April, R_e continued to fluctuate around 1, which is consistent with N_e remaining relatively stable throughout March. Additionally, while there was underreporting at the start of the fifth wave, as confirmed by a decrease in the relative case detection rate, our estimation of prevalence and incidence suggests that more substantial under-ascertainment occurred since March, despite the inclusion of RAT-positive cases.

“Interestingly, from March to mid-April, R_e continued to fluctuate around 1 in comparison to R_t , which was less than 1, indicating a slower decline of the outbreak than anticipated.”

“The early exponential increase in N_e stabilized from late January to early February, coinciding with the decrease in R_e . However, N_e rebounded in early February with a sharp increase in late February 2022, peaking around 9 March 2022, and remained relatively stable throughout March (Fig. 3b).”

“According to our estimates of prevalence and incidence, the epidemic peaked on the week from 28 Feb to 6 March 2022 (Table 1 and Fig. 4). Despite the inclusion of RAT-positive cases, more substantial under-ascertainment occurred since March, with the exception of under-ascertainment at the start of the fifth wave, as evidenced by a decrease in the relative case detection rate (Fig. 3c).”

Reviewer 1, comment 6. On that note, the formula used by Bedford estimate prevalence/incidence from N_e is actually from Koelle and Rasmussen (2012; <https://www.ncbi.nlm.nih.gov/pmc/articles/PMC3306638/>). I am unsure if this is the right approach to compute incidence from N_e because this was formulated under assumption that disease dynamics were at their endemic equilibrium, which is certainly not the case here under exponential growth).

Response to R1C6. We thank the reviewer for bringing to our attention the correct implementation and citation. To better capture these complex dynamics under exponential growth, we followed a discrete generation model with arbitrary offspring distribution and changing population size [5] to estimate prevalence from effective population size. Due to the superspreading dynamics of SARS-CoV-2 [6, 7], a flexible offspring distribution, called the negative binomial, was assumed, which has a dispersion parameter k that controls its shape. The updated formulas used in this analysis are listed in lines 470-477:

“Given the complex dynamics of the fifth wave in Hong Kong, we estimated point prevalence (I) from N_e at generation time (τ), following a discrete generation model with arbitrary offspring distribution and changing population size. Due to the superspreading dynamics at SARS-CoV-2 [6, 7], the negative binomial offspring distribution was assumed, which has a dispersion parameter (k) that controls its shape. Point prevalence (I) can be calculated using the formula below:

$$I = \frac{N_e}{\tau} * (\sigma^2/R + R - 1)$$

subject to:

$$\sigma^2 = R + R^2/k$$

Reviewer 1, comment 7. The authors claimed that “at least four to five instances of hamster-to-human transmission were observed according to the Bayesian maximum clade credibility tree and the maximum likelihood phylogeny, respectively (Supp fig.2).” While there have been likely hamsters-to-human transmission events reported previously, I am unsure if the trees presented by authors suggest that. At best, it shows that there is a shared common ancestor between the sampled viruses in humans and hamsters but the analyses neither show the directionality of transmissions nor the likely host species of the common ancestor. I actually find this point to be rather distracting as I read through the paper and would advise the authors to take this part out.

Response to R1C7. We agree with the reviewer and removed these lines.

Reviewer #2 (Remarks to the Author):

The approach of this paper – to examine and track underlying drivers of transmission behind COVID-19 variants (particularly omicron sub-variants) in Hong Kong, which presents a unique mix of age-stratified susceptibility given its earlier measures, is interesting and of potentially wider importance. However, I have major concerns about the methodology, interpretation and validation of the results and in particular the conclusions drawn from the estimates of R_e and R_t . These issues need to be resolved for this work to be convincing, especially as the central claims in this paper follow from the analyses in question.

Specifically, not enough is done to confirm the accuracy of the R_e of 9.5, which appears to be the basis of the key conclusion of the paper about under-reporting. There are several issues:

Reviewer 2, comment 1. The birth-death skyline is well known to suffer **biases from sampling proportions** (as all the parameters in this model are not identifiable). While checks against the prior on R_e are important and provided – similar examination of the sampling proportion prior choice and sensitivity tests are needed. This is especially important given that it can have a large impact on R_e .

Response to R2C1. We thank the reviewer for these critical comments. We now include sensitivity tests for different sampling proportion priors, showing that unreliable priors (previously defaulted priors) could lead to convergent high sampling proportions, thereby overestimating R_e unexpectedly (Supplementary Note and Supplementary Fig. 9). For further details please see response to Reviewer 1 comment 3.

Reviewer 2, comment 2. While R_e is drawn from BA.2 sequences it is compared to an R_t inferred from all local cases (and if it is not then results concerning potential co-circulating lineages later in the time series seem odd). The large R_e occurs around Jan 10-17 but there are also few sequences and cases of BA.2 in this period. This is when reproduction numbers are most difficult to infer, especially resurgences. Consequently, validation of these estimates would help e.g., by showing how well R_e or R_t can sequentially reproduce its data e.g., <https://www.sciencedirect.com/science/article/pii/S0022249606000058> and <https://sites.stat.washington.edu/raftery/Research/PDF/Gneiting2007jrssb.pdf>

Response to R2C2. We appreciate the suggestion made by the reviewer and have taken it into consideration in our analysis. We compared the ability of R_e and R_t to sequentially reproduce the dynamics of the initial period of the fifth wave (10–24 January) in Hong Kong using probabilistic forecasting metrics. We used accumulative one-step-ahead prediction error (APE) algorithms [8], and followed methods described in [9] and [10] to convert R_e and R_t to incidence without accounting for superspreading and compared with reported local cases. Our results showed that R_t was able to better reproduce the dynamics of the reported cases on 10–24 January 2022 compared to R_e . R_t is based on all local cases, while R_e is derived from BA.2.2 sequences (Supplementary Note and Supplementary Fig. 4). However, extensive R_e estimations based on sensitivity analysis, backed by strong epidemiological records indicating the virus entered the community as early as 10 January, suggest that R_e increased prior to R_t , revealing undiscovered community transmission chains during the initial period of the fifth wave. We clarified this in lines 164-167 in the manuscript and lines 22-35 in Supplementary Note:

“Early in outbreaks, it is often difficult to estimate effective reproduction numbers accurately. Herein, we investigated how well R_e and R_t can sequentially reproduce

the dynamics of the initial fifth wave of SARS-CoV-2 (10-24 January) in Hong Kong using probabilistic forecasting metrics. Following [9] and [10], the number of new infections at time t could be roughly (superspreading is not considered) inferred by multiplying the reproduction number by the total infectiousness of infected individuals at time t , given by the sum of infection incidence up to time step $t - 1$, weighted by the infectivity function based on the gamma probability distribution (w_t) of an individual's infectivity profile once infected [10]. Based on estimated incidence and observed reported cases, the forecasting metrics of absolute error, calibration, and sharpness were calculated implemented in the R package “scoringutils” [11] [12]. As evidenced by lower absolute error, larger calibration, and lower sharpness in most time periods (Supplementary Fig. 4), R_t was able to better reproduce the dynamics of reported cases in 10-24 January 2022 than R_e . Due to the fact that R_e is derived from BA.2.2 sequences, while R_t is derived from all local cases, this result is expected.”

“Although R_t was able to reproduce the dynamics of reported cases better than R_e during 10-24 January 2022 (Supplementary Note and Supplementary Fig. 4), the increase in R_e , as well as the potential for uncaptured transmission at the hotel, suggest that some community transmission chains was not identified.”

Reviewer 2, comment 3. There are many potential explanations for a large initial R_e that could also explain this result e.g., <https://onlinelibrary.wiley.com/doi/10.1002/sim.4174> but not all of them are ruled out or discussed. Strengthening the rejection of these other hypotheses would be useful if the claim that the difference in R_e and R_t is largely due to sampling.

Response to R2C3. As suggested by the reviewer, in lines 284-297, we discussed potential factors that could lead to overestimations early in outbreaks.

“In the early stages of an outbreak, the reproductive number is commonly overestimated due to many factors[13], such as incorrectly accounting for imported cases and subpopulations with higher transmission rates. In this study, the first community case (Case B) was detected and imported cases were excluded via extensive contact tracing. Whether the intrinsic transmission rate of SARS-CoV-2 is higher in particular subpopulations (e.g. children and/or the elderly) in Hong Kong is unknown, and whether this could result in overly high estimates of reproductive numbers require further study. In addition, our previous study[14], using comprehensive simulation analysis, showed our approach for R_t estimation would tend to underestimate R_t when R_t is increasing, while overestimate R_t when R_t is decreasing, but could still provide the correct direction of change of R_t . In our study, we have discussed how variable sampling of sequences throughout the outbreak could overestimate R_e in the BDSKY model if unreliable prior assumptions of sampling proportions are used (Supplementary Note and Supplementary Figs. 9 and 10). These biases could account for the difference in R_e and R_t and have an impact on interpreting the dynamics of the fifth wave in Hong Kong.”

Reviewer 2, comment 4. However, even if these confounders are not important the claim itself only works if a significant change in reporting rates occurs. There is such a change from PCR to RAT but this occurs much later.

As <https://academic.oup.com/aje/article/178/9/1505/89262> and <https://www.nature.com/articles/s43588-022-00313-1> show, much of the information about R_t is preserved in periods of relatively constant rate. These points need to be discussed and qualified especially with regards to the timing of effects.

Response to R2C4. We appreciate the reviewer's insight. Our R_t estimation approach, as described in Cori et al. [9], uses a smoothing method with a constant transmissibility assumption over the time period $[t-\tau+1, t]$, where τ is the smoothing window parameter. However, the choice of the smoothing window size can affect both the temporal and quantitative accuracy of estimates. Larger windows can provide more stable R_t estimates by leveraging information from multiple time points, but can also obscure epidemiologically meaningful changes in R_t . To strike a balance, we have set $\tau=14$ in our analysis. Our simulation studies have shown that our approach tends to underestimate R_t during periods of decreasing transmission, but still correctly captures the direction of change of R_t . This was discussed below.

Methods lines 464-468 "As Cori et al. [9] and Parag et al. [15] show, R_t is assumed constant over a time period $[t-\tau+1, t]$, where τ is the smoothing parameter. In this study, we take $\tau=14$, to avoid unstable estimates for time-varying reproductive number. Correspondingly, the estimated R_t would need a few days to move to its true value, but still provide the correct direction of change [14]."

Discussion lines 290-293 "In addition, our previous study[14], using comprehensive simulation analysis, showed our approach for R_t estimation would tend to underestimate R_t when R_t is increasing, while overestimate R_t when R_t is decreasing, but could still provide the correct direction of change of R_t ."

Reviewer 2, comment 5. The rise in N_e occurs in line with the major fall in R_e – how can this be explained? If N_e is proportional to BA.2 infections and its growth rate increases, then one would not expect R_e to fall to 1 or below at that time (even if generation times change).

Response to R2C5. We have revised this section based on our revised estimates of R_e . As indicated by the reviewer, we found that the rise in N_e stabilised from 27 Jan immediately after the fall in R_e . For further details please see response to Reviewer 1 comments 3 and 5.

Lines 192-196: "The early exponential increase in N_e stabilized from late January to early February, coinciding with the decrease in R_e . However, N_e rebounded in early February with a sharp increase in late February 2022, peaking around 9 March 2022, and remained relatively stable throughout March (Fig. 3b)."

Reviewer 2, comment 6. It also appears that the imposition of the school face-to-face intervention correlated with a fall in R_t but a fall in R_e had already occurred notably earlier. This would also need to be explained given that timing of events is often included in the results.

Response to R2C6. Per revised estimates the sharp decline in R_e commences after the intervention, agreeing with R_t estimates. We have also highlighted that while R_t is measured daily, R_e is a weekly estimate. Please see response to Reviewer 1 comments 3.

Reviewer 2, comment 7. The sampling of sequences in constructing R_e is also important – what schemes were used and were any validation tests across different sequence undersampling schemes considered? This choice can strongly influence R_e .

Response to R2C7. Thanks you for bringing up the importance of the sampling in constructing R_e . We employed two commonly used sampling methods, uniform and proportional, recommended by the WHO for practical use in various settings. We used these schemes to create three datasets with sample sizes of 262 (uniform, 20 sequences per week), 502 (uniform, 40 sequences per week), 897 (proportional) (Supplementary Fig. 11). On the basis of three datasets, we show overlapping confidence intervals and no potential bias in estimations of R_e (Supplementary Note and Supplementary Fig. 10).

Reviewer 2, comment 8. Are there differences in the definitions of R_e and R_t ? The instantaneous vs effective terminology is used but not explained

Response to R2C8. Per reviewer’s suggestion, we clarified that R_e was estimated using the BDSKY model based on sampled sequences and sample dates over 16 time intervals (corresponding to 16 weeks). On the other hand, R_t was calculated daily using the reported number of local infections on each day. Please refer to the results section (lines 154-158, 161-164) in the revised manuscript for more details.

“To reveal changes in the spread of BA.2.2 in Hong Kong over time, we used a Bayesian birth-death skyline model that explicitly estimates the rate of transmission, recovery, and sampling, enabling a direct inference of the effective reproductive number (R_e) based on sampled sequences and sample dates over 16 time intervals, roughly corresponding to weeks between 3 January and 26 April.”

“The instantaneous effective reproduction number (R_t), estimated from the number of local infections reported per day, increased gradually from 1 (HPD, 0.6–2.2) on 12 January and peaked at 5.2 (HPD, 3.9–7.7) on 20 January.”

Reviewer #3 (Remarks to the Author):

The paper describes the epidemiology of a large wave of predominantly Omicron BA.2 cases, in a relatively immunologically naïve population. This resulted in high mortality, despite implementation of multiple non-pharmaceutical interventions. The results show the importance of keeping track of unbiased case rates and effective reproduction numbers in near-real time, to avoid ill-informed policy decisions. A number of important lessons can be learned from this study, including the need for timely sequencing of confirmed cases and subsequent analysis of genomic and epidemiologic data - it's clear that data were not available (or analysed) in time for crucial public health decisions to be made.

The methods seem generally sound based on my understanding of the field, although I do have a number of questions and comments:

Reviewer 3, comment 1. When were the samples sequenced and made available for analysis relative to the outbreak? What explains the highly variable proportion of notified cases sequenced throughout the outbreak, and why were so few sequenced from mid-February onwards? It would be helpful to understand more about the application of genomic surveillance in Hong Kong and how that changed over time, and discuss the implications of such highly variable sampling of sequences on estimates of R_e and N_e (i.e. could it have biased estimates?).

Response to R3C1. We appreciate the insightful comment. Our analysis of SARS-CoV-2 genomes submitted to GISAID between January and April 2022 showed that routine sequencing in Hong Kong was performed within a timeframe of approximately two weeks, with a mean number of 45.4 sequences submitted per week (median: 32; range: 1–194; Supplementary fig. 8). However, the overall sampling proportion declined from about 30% to less than 1% since February (Supplementary Table 1). As described in lines 298-308, these findings have implications for the estimation of R_e and N_e , as the highly variable sampling of sequences throughout the outbreak could result in an overestimation of R_e , particularly if unreliable priors are used in the BDSKY model (Supplementary note and Supplementary figs. 9 and 10).

“Furthermore, GISADI sequence submission records between January and April 2022 show that sequencing in Hong Kong was typically completed within two weeks. However, the mean number of sequences submitted with a delay of less than two weeks was only 45 per week (median: 32; range: 1-194; Supplementary Fig. 8). This was inadequate considering the hundreds of confirmed daily case counts since February, when the total sampling proportion declined from ~30% to less than 1% (Supplementary Table 1). Underestimation of R_e could occur if the sampling proportion is small, as observed since February, which failed to capture the entire genetic diversity revealed through N_e . When RAT positive cases were included in public reporting from 26 February, a further sharp spike in N_e followed. This suggests that BA2.2 sublineages that circulated cryptically were better captured. These observations indicate the timeliness and quantity of genomic surveillance in Hong Kong should be improved.”

Reviewer 3, comment 2. In Figure 1 I found the situation concerning RATs confusing - there is a marker on March 7th that says when the online platform for reporting RATs was introduced but there are a large number of RATs reported after Feb 26th when the confirmed cases included RATs from private hospitals and laboratories. Did the system

change to include RATs from sources other than private hospitals and laboratories on March the 7th? Some clarity in the figure and/or figure legend would make the situation clearer concerning RATs. Also I suggest you say that the spike on March the 9th includes cases from the 8th to make it clearer that this is an artifact in the data.

Response to R3C2. Per reviewer’s suggestion, we changed the markers, bolded the “RATs” in Figure 1, updated the legend and rewrote the corresponding text (lines 35-39) as follows:

“As established systems for testing became overwhelmed, the Centre for Health Protection (CHP) pivoted to include positive rapid antigen test (RAT) cases from private hospitals and laboratories in official case counts since 26 February (Fig. 1), rather than only recognising PCR-positives confirmed by Government reference laboratories (Fig. 1). A self-declaration system for positive RAT reporting was launched on 7 March.”

“Rapid antigen test positive cases reported on 9 March include cases from both 8 and 9 March.”

Reviewer 3, comment 3. The effective reproductive rate (R_e) and the instantaneous effective reproduction rate (R_t) could be more clearly defined earlier in the text, and described more clearly in the methods. Note in epidemiological terms they are numbers and not true rates and should really be named as such.

If I understand it correctly, R_e is determined from the birth-death model using sampled sequence data and (presumably), sample dates, whereas R_t is estimated from reported incidence data, after imputing infection dates.

I recommend defining R_e and R_t briefly in the results section, as well as in more detail in the methods section. For example in the results section they could be defined as R_e = effective reproduction number estimated using sampled sequences and sample dates over XX time intervals (i.e. indicate what time grid was used in bdskytools), and R_t = instantaneous effective reproduction number calculated daily using the estimated number of infections on each day.

Response to R3C3. We thank the reviewer for this suggestion. We revised our text to better reflect the definitions of R_e and R_t , including using number instead of rate. In the result section (lines 154-158, 161-164), R_e is defined as “To reveal changes in the spread of BA.2.2 in Hong Kong over time, we used a Bayesian birth-death skyline model that explicitly estimates the rate of transmission, recovery, and sampling, enabling a direct inference of the effective reproductive number (R_e) based on sampled sequences and sample dates over 16 time intervals, roughly corresponding to weeks between 3 January and 26 April.” And R_t is defined as “The instantaneous effective reproduction number (R_t) estimated daily from the number of local infections reported per day, increased gradually from 1 (HPD, 0.6–2.2) on 12 January and peaked at 5.2 (HPD, 3.9–7.7) on 20 January.”

Reviewer 3, comment 4. Page 17: The last two equations seem to be mislabelled.

Shouldn't $P(\text{tested})$ be $1 - (1 - 1/\text{pop})^{nt}$ and $P(\text{infected})$ be $c * N_e / \text{pop}$?

Assuming this is correct, how is c defined in biological terms? Presumably it represents the number of true cases per effective population 'unit'?

Response to R3C4. We apologize for the mislabelling of the equations and lack of clarity in the definition of c . To address this concern, we have revised the text to clarify that c represents the number of true cases per effective population unit and added a conversion formula that shows how $P(\text{infected})$ is calculated. The revised text reads:

“

$$P_t(\text{infected}) = \frac{pop_{\text{infected}}}{pop} = \frac{c * N_e}{pop}$$

$$P_t(\text{tested}) = (1 - (1 - \frac{1}{pop})^{n_t})$$

where pop_{infected} is the number of infections in the population which can be simplified as a constant factor (c , which represents the number of true cases per effective population 'unit') times N_e due to their linear correlation. pop is population size (7.4 million) in Hong Kong, r_{pos} denotes the positivity rate of the PCR tests conducted and n_t represents the number of tests conducted.”

Reviewer 3, minor comments

Page 3: "peak of 3.5 per 100,000 people" please give units (per day?)

Yes, per day. Added.

Page 6: "...lineage introduced by from imported..." should be "lineage introduced by imported..."

Modified as indicated.

Page 6: "Delta cases detected in the community between 15 January and late March" I can't see any evidence of Delta cases in March in Figure 1 or Supp Figure 2?

We apologize for the confusion. The latest complete Delta AY.127 genome sequence that was sampled in Hong Kong was on 13 February 2022. In late March, there was a report of an incomplete Delta AY.127 sequence, however it was not available for analysis. We have corrected the date in the text, changing "late March" to "13 February" in lines 100-103: "Delta cases detected in the community between 15 January and 13 February formed a single monophyletic lineage introduced by imported pet hamsters and first reported on 17 January 2022 (Supplementary Fig.2)"

Page 6: "152 detections did not lead to onward transmission" suggest "152 detections did not lead to detectable onward transmission". Important given the low proportion of cases sequenced.

Modified.

Page 8: "...of viruses with spike I1221T viruses was..." repetition of "viruses"

Deleted.

Page 8: "...as indicated by the light blue dashed line" should be red dashed line.

Modified as indicated.

Figure 3a - y axis should be labelled R_e/R_t and it should be made clear in the text and the legend that R_t refers to all cases, not just BA.2.2 (assuming that is the case).

Modified as suggested. Please also see response to comment 3.

“(a) The effective reproduction number (R_e) based on BA.2.2 sequences and the instantaneous effective reproduction number (R_t) based on daily reported number of local cases”

Page 13: "...indicated a 15–73% under detection, whereas a high dispersion ($k = 0.05$) showed a 2–4-fold under detection of cases". Suggest you provide underreporting rates in the text, as in Table 1, or relative underreporting rates i.e. be consistent in either % under detection or X fold relative under detection.

Modified as suggested.

References

1. Sanderson, T., *Taxonium, a web-based tool for exploring large phylogenetic trees*. Elife, 2022. **11**.
2. Leung, K., J.T. Wu, and G.M. Leung, *Real-time tracking and prediction of COVID-19 infection using digital proxies of population mobility and mixing*. Nat Commun, 2021. **12**(1): p. 1501.
3. Lam, H.K.W. and M.G. Bell, *Advanced modeling for transit operations and service planning*. 2003.
4. Stadler, T., et al., *Birth-death skyline plot reveals temporal changes of epidemic spread in HIV and hepatitis C virus (HCV)*. Proc Natl Acad Sci U S A, 2013. **110**(1): p. 228-33.
5. Fraser, C. and L.M. Li, *Coalescent models for populations with time-varying population sizes and arbitrary offspring distributions*. bioRxiv, 2017: p. 131730.
6. Riou, J. and C.L. Althaus, *Pattern of early human-to-human transmission of Wuhan 2019 novel coronavirus (2019-nCoV), December 2019 to January 2020*. Euro Surveill, 2020. **25**(4).
7. Adam, D.C., et al., *Clustering and superspreading potential of SARS-CoV-2 infections in Hong Kong*. Nat Med, 2020. **26**(11): p. 1714-1719.
8. Wagenmakers, E.-J., P. Grünwald, and M. Steyvers, *Accumulative prediction error and the selection of time series models*. Journal of Mathematical Psychology, 2006. **50**(2): p. 149-166.
9. Cori, A., et al., *A new framework and software to estimate time-varying reproduction numbers during epidemics*. Am J Epidemiol, 2013. **178**(9): p. 1505-12.
10. Parag, K.V., *Improved estimation of time-varying reproduction numbers at low case incidence and between epidemic waves*. PLoS Comput Biol, 2021. **17**(9): p. e1009347.
11. Bosse, N., S. Abbott, and F.S. EpiForecasts, *Scoringutils: Utilities for scoring and assessing predictions*. 2020.
12. Jordan, A., F. Krüger, and S. Lerch, *Evaluating Probabilistic Forecasts with scoringRules*. Journal of Statistical Software, 2019. **90**(12): p. 1 - 37.
13. Mercer, G.N., K. Glass, and N.G. Becker, *Effective reproduction numbers are commonly overestimated early in a disease outbreak*. Stat Med, 2011. **30**(9): p. 984-94.

14. Tsang, T.K., et al., *Accounting for Imported Cases in Estimating the Time-Varying Reproductive Number of Coronavirus Disease 2019 in Hong Kong*. J Infect Dis, 2021. **224**(5): p. 783-787.
15. Parag, K.V., C.A. Donnelly, and A.E. Zarebski, *Quantifying the information in noisy epidemic curves*. Nature Computational Science, 2022. **2**(9): p. 584-594.

REVIEWER COMMENTS

Reviewer #1 (Remarks to the Author):

The authors have adequately addressed all of my concerns and comments. Thank you. I have no other outstanding concerns.

Reviewer #2 (Remarks to the Author):

I thank the authors for their comprehensive responses and do believe they have adequately addressed several of my queries and improved the manuscript. However, a couple key issues are not yet resolved and are important for some stated conclusions and insights this paper provides. Particularly, while the updated R_e no longer shows extremely large values and is more consistent with R_t , the use of these estimates as evidence in of itself for under-ascertainment seems, currently, shaky.

1. Supplement Fig 9 indicates that across January the analysis is least robust to sampling choices and Table S1 shows a sampling proportion of above 100% for 17-23 January. The prior minimum of 0.3 is set as the proportion from 1-16 January. Can these choices be better justified given how sensitive estimates are in this period?
2. A central claim of this paper relates to “ R_e increased earlier showing potential for R_t estimation methods that are dependent on case reporting to miss key dynamic information during the outbreak.” This could be true but there is such strong uncertainty in the R_e and R_t estimates in that period, with major overlap of their credible intervals, that the analysis does not appear to sufficiently support the claim. What would $P(R_e > 1)$ and $P(R_t > 1)$ look like (as this somewhat includes the credible intervals)?
3. “In addition, our previous study [14], using comprehensive simulation analysis, showed our approach for R_t estimation would tend to underestimate R_t when R_t is increasing, while overestimate R_t when R_t is decreasing, but could still provide the correct direction of change of R_t .” Isn’t this also confounding the likely smaller increase in R_t initially? Further, both R_e and R_t are clearly above 1 so qualitatively similar signals are present.
4. R_t appears to be derived from cases that include other variants which could have lower intrinsic transmissibility. It may be better to compare R_e to R_t from just the local cases that include the variant of interest.
5. Across Feb 9-21 R_t is significantly above 1 while R_e is roughly 1. These would indicate differing views of the epidemic. Can this be explained?
6. “We have also highlighted that while R_t is measured daily, R_e is a weekly estimate.” But isn’t R_t smoothed with a 2-week long window?
7. I agree with comparing projections from R_t to reported cases but projections from R_e should be compared to the sequence time series. If those projections reproduce this data (can show visually) then this indicates well-calibrated R_e estimates. Computing scores of those projections at different settings e.g., under various sampling proportion priors would then reinforce that the estimates chosen are optimal in some sense.

In short, I don’t think a stated aim – “This study demonstrates that relying on R_t estimation methods dependent on case reporting can lead to misinformed epidemic response planning and highlights the need for research to improve near real-time epidemic growth estimates.” – is yet achieved. An alternative could be to excise or substantially reduce the claims depending on Fig 3 and then only do additional analyses to fill any remaining gaps.

Responses to Reviewer Comments

Reviewer #2 (Remarks to the Author):

I thank the authors for their comprehensive responses and do believe they have adequately addressed several of my queries and improved the manuscript. However, a couple key issues are not yet resolved and are important for some stated conclusions and insights this paper provides. Particularly, while the updated R_e no longer shows extremely large values and is more consistent with R_t , the use of these estimates as evidence in of itself for under-ascertainment seems, currently, shaky.

Response to Reviewer 2's general remarks We appreciate these constructive comments to improve the manuscript and have addressed each of the comments below and made corresponding changes in the revised manuscript.

Reviewer 2, comment 1. Supplement Fig 9 indicates that across January the analysis is least robust to sampling choices and Table S1 shows a sampling proportion of above 100% for 17-23 January. The prior minimum of 0.3 is set as the proportion from 1-16 January. Can these choices be better justified given how sensitive estimates are in this period?

Response to R2C1. We apologize for any confusion caused. We previously listed the sequencing proportions in Table S1, which were 100% during the specified period. However, we have now updated Table S1 to include the subsampling proportions used to estimate R_e under uniform and proportional sampling schemes. Furthermore, we conducted a sensitivity analysis under different sampling proportion priors from 10–17 January (as shown in Figure 1 below), which revealed that setting upper and lower bounds for the sampling proportion prior could lead to higher R_e estimates between 10 and 17 January (Lines 467-469). While the overall sampling proportion was limited to a low level based on epidemiological priors, the sensitivity analysis results were not as pronounced as those shown in Supplementary Fig. 9. Based on our epidemiological data and the slight changes of R_e observed in the sensitivity analysis, we have decided to retain the prior minimum of 0.3 and allow it to iterate from 0.3 to 1 in the MCMC framework.

“the upper and lower bounds of the sampling proportion prior could lead to a higher R_e between 10 and 17 January.”

Figure 1. The effective reproductive number (R_e) over time under different sampling proportion assumptions.

Reviewer 2, comment 2. A central claim of this paper relates to “ R_e increased earlier showing potential for R_t estimation methods that are dependent on case reporting to miss key dynamic information during the outbreak.” This could be true but there is such strong uncertainty in the R_e and R_t estimates in that period, with major overlap of their credible intervals, that the analysis does not appear to sufficiently support the claim. What would $P(R_e > 1)$ and $P(R_t > 1)$ look like (as this somewhat includes the credible intervals)?

Response to R2C2. We acknowledge the reviewer’s comment and agree that the evidence is insufficient to support the claim that relying solely on R_t estimation could miss crucial information during the outbreak. Although R_e exceeded 1 earlier than R_t from 11–17 January 2022, there is a overlap between the credible intervals, and the signals are similar (Table 1 below and Fig. 3a in the manuscript). Moreover, the estimation of R_t requires a few days to reach its true value, as discussed in the Methods and Discussion sections. Therefore, we have revised the statement in the Abstract to reflect this:

“Discordant inferences based on genomic and epidemiological data underscore the need to improve near real-time epidemic growth estimates by combining multiple disparate data sources to better inform outbreak response policy.”

Table 1. Probability of $R_e > 1$ and $R_t > 1$ from 11-16 January 2022.

Date	P ($R_e > 1$)	P ($R_t > 1$)
2022-01-11	0.515	0.911
2022-01-12	0.905	0.684
2022-01-13	0.985	0.457
2022-01-14	0.985	0.365
2022-01-15	0.990	0.474
2022-01-16	0.990	0.681
2022-01-17	0.990	0.901

Reviewer 2, comment 3. “In addition, our previous study [14], using comprehensive simulation analysis, showed our approach for R_t estimation would tend to underestimate R_t when R_t is increasing, while overestimate R_t when R_t is decreasing, but could still provide the correct direction of change of R_t .” Isn’t this also confounding the likely smaller increase in R_t initially? Further, both R_e and R_t are clearly above 1 so qualitatively similar signals are present.

Response to R2C3. Please see response to Reviewer 2 comments 2.

Reviewer 2, comment 4. R_t appears to be derived from cases that include other variants which could have lower intrinsic transmissibility. It may be better to compare R_e to R_t from just the local cases that include the variant of interest.

Response to R2C4. The local cases for the variant of interest are not available, so we cannot estimate R_t excluding other variants.

Reviewer 2, comment 5. Across Feb 9–21 R_t is significantly above 1 while R_e is roughly 1. These would indicate differing views of the epidemic. Can this be explained?

Response to R2C5. We provided two possible explanations in lines 177-181: Either there were other lineages co-circulating (e.g., BA.2.10.1) or HK-BA.2.2 was under-sequenced which lead to an under estimation of R_e during this period.

“There was a similar dynamic pattern in R_t , but from 7–28 February, R_t remained above 2 with a slight decrease, significantly higher than R_e which fluctuated around 1. Higher R_t may reflect the inclusion of cases caused by other co-circulating lineages (e.g., BA.2.10.1) (Fig. 2c) in which R_e has not been considered, or it may reflect under-sequencing of HK-BA.2.2 during this period.”

Reviewer 2, comment 6. “We have also highlighted that while R_t is measured daily, R_e is a weekly estimate.” But isn’t R_t smoothed with a 2-week long window?

Response to R2C6. We apologise for the oversight as this statement in our response letter is incorrect. This incorrect statement was not included in the manuscript.

Reviewer 2, comment 7. I agree with comparing projections from R_t to reported cases but projections from R_e should be compared to the sequence time series. If those projections reproduce this data (can show visually) then this indicates well-calibrated R_e estimates. Computing scores of those projections at different settings e.g., under various sampling proportion priors would then reinforce that the estimates chosen are optimal in some sense.

Response to R2C7. Per reviewer’s request, we compare the projections from R_e with the sequence time series under various sampling proportion priors. The results showed large differences between estimated incidence from R_e under the low sampling proportion prior (0.1) and sequencing time series (Supp fig. 4). Overall, our results indicate that the projections from R_t and R_e (using sampling proportion > 0.3) were able to reasonably reproduce the incidence data, suggesting well-calibrated transmissibility estimations in this study. This is further reinforced by the low absolute error, high calibration, and low sharpness of the projections (Supp fig. 4; Supp information, lines 46-55).

“Specifically, for R_e , we used sequencing time series under various sampling proportion priors to replace observed reported cases for this projection. As the sampling proportion is over 100% for 17-23 January, we assumed that all reported cases were sequenced. Sampling proportion priors with 0.1, 0.3, 0.5 and 0.7 for 10-16 January were tested. Our results showed large differences between estimated incidence from R_e under low sampling proportion priors (0.1) and sequencing time series (Supp fig. 4). Nevertheless, those projections from R_t and R_e (sampling proportion > 0.3) were able to reproduce this incidence data to some extent, suggesting well-calibrated transmissibility estimations in this study, marked by low absolute error, high calibration, and low sharpness (Supp fig. 4).”

Reviewer 2, comment 8. In short, I don’t think a stated aim – “This study demonstrates that relying on R_t estimation methods dependent on case reporting can lead to misinformed epidemic response planning and highlights the need for research to improve near real-time epidemic growth estimates.” – is yet achieved. An alternative could be to excise or substantially reduce the claims depending on Fig 3 and then only do additional analyses to fill any remaining gaps.

Response to R2C8. We thank the reviewers for the comprehensive review. As indicated above, we have rewritten this statement in the abstract.

“Discordant inferences based on genomic and epidemiological data underscore the need to improve near real-time epidemic growth estimates by combining multiple disparate data sources to better inform outbreak response policy.”

REVIEWERS' COMMENTS

Reviewer #2 (Remarks to the Author):

I thank the reviewers for performing the additional analyses and am largely happy with the responses. Supplementary Fig 4 isn't as definitive as I would like in terms of indicating good one-step-ahead predictions, but it does help and I understand the issues of dealing with real data and so require no further revisions.

My final very minor comment is to double check the text below.

“There was a similar dynamic pattern in R_t , but from 7–28 February, R_t remained above 2 with a slight decrease, significantly higher than R_e which fluctuated around 1. Higher R_t may reflect the inclusion of cases caused by other co-circulating lineages (e.g., BA.2.10.1) (Fig. 2c) in which R_e has not been considered, or it may reflect under-sequencing of HK-BA.2.2 during this period.”

If attributed to other lineages mixed into R_t then these have to either have much larger intrinsic transmissibility or moderately higher transmissibility and be dominant in proportion I think in order for R_t to be that high.

Responses to Reviewer Comments

Reviewer #2 (Remarks to the Author):

I thank the reviewers for performing the additional analyses and am largely happy with the responses. Supplementary Fig 4 isn't as definitive as I would like in terms of indicating good one-step-ahead predictions, but it does help and I understand the issues of dealing with real data and so require no further revisions.

My final very minor comment is to double check the text below.

“There was a similar dynamic pattern in R_t , but from 7–28 February, R_t remained above 2 with a slight decrease, significantly higher than R_e which fluctuated around 1. Higher R_t may reflect the inclusion of cases caused by other co-circulating lineages (e.g., BA.2.10.1) (Fig. 2c) in which R_e has not been considered, or it may reflect under-sequencing of HK-BA.2.2 during this period.”

If attributed to other lineages mixed into R_t then these have to either have much larger intrinsic transmissibility or moderately higher transmissibility and be dominant in proportion I think in order for R_t to be that high.

Response to Reviewer 2's Comment We appreciate the insightful comment to help us better interpret the differences between R_e and R_t . In the revised manuscript, we rule out the possibility of BA.2.10.1 contributing higher R_t from 7–28 February since it was not the dominant lineage during this period. Please see lines 192–193 in the final manuscript.